# Disentangled and Self-Explainable Node Representation Learning

**Simone Piaggesi**                                          *simone.piaggesi@di.unipi.it*
*University of Pisa, Pisa, Italy*

**André Panisson**                                          *andre.panisson@centai.eu*
*CENTAI Institute, Turin, Italy*

**Megha Khosla**                                          *M.Khosla@tudelft.nl*
*Delft University of Technology, Delft, Netherlands*

**Reviewed on OpenReview:** *https://openreview.net/forum?id=s51TQ8Eg1e*

## Abstract

Node embeddings are low-dimensional vectors that capture node properties, typically learned through unsupervised structural similarity objectives or supervised tasks. While recent efforts have focused on post-hoc explanations for graph models, intrinsic interpretability in *unsupervised* node embeddings remains largely underexplored. To bridge this gap, we introduce DISENE (**Di**sentangled and **Se**lf-Explainable **N**ode **E**mbedding), a framework that learns self-explainable node representations in an unsupervised fashion. By leveraging disentangled representation learning, DISENE ensures that each embedding dimension corresponds to a distinct topological substructure of the graph, thus offering clear, dimension-wise interpretability. We introduce new objective functions grounded in principled desiderata, jointly optimizing for structural fidelity, disentanglement, and human interpretability. Additionally, we propose several new metrics to evaluate representation quality and human interpretability. Extensive experiments on multiple benchmark datasets demonstrate that DISENE not only preserves the underlying graph structure but also provides transparent, human-understandable explanations for each embedding dimension.

## 1 Introduction

Self-supervised and unsupervised node representation learning (Hamilton, 2020) provide a powerful toolkit for extracting meaningful insights from complex networks, making them essential in modern AI and machine learning applications for network analysis (Ding et al., 2024). These methods offer flexible and efficient ways to analyze high-dimensional networks by transforming them into low-dimensional vector spaces. This transformation enables dimensionality reduction, automatic feature extraction, and the deployment of standard machine learning algorithms for tasks such as node classification, clustering, and link prediction (Khosla et al., 2021). Moreover, unsupervised node representations, or embeddings, enable visualization of complex networks and can be transferred across similar networks, enhancing understanding and predictive power in fields ranging from social networks to biological systems. Despite their widespread utility, these approaches often face substantial challenges in terms of interpretability, typically relying on complex and post-hoc techniques to understand the latent information encoded within the embeddings (Piaggesi et al., 2024; Idahl et al., 2019; Gogoglou et al., 2019). This limitation raises a critical question: *What information do these embeddings encode?*

Despite a large body of research on explainable GNN models, embedding methods–the fundamental building blocks of graph-based systems–have received comparatively little attention. Most existing attempts to explain embeddings are predominantly post-hoc (Piaggesi et al., 2024; Gogoglou et al., 2019; Khoshraftar

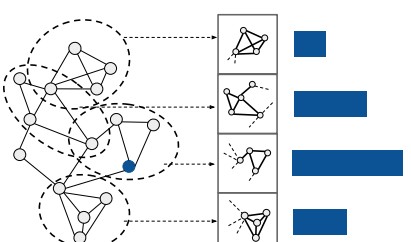 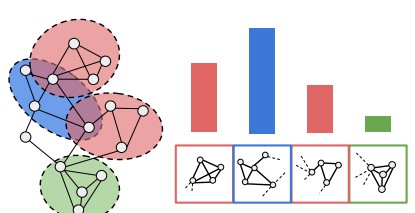 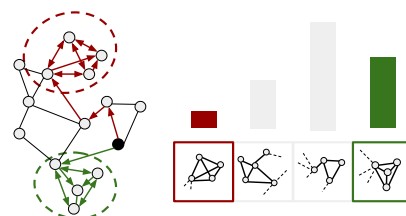

Figure 1: DISENE generates dimension-wise disentangled representations in which each embedding dimension is mapped to a mesoscale substructure in the input graph. The vector represents the embedding for the node marked in blue and the bars depict feature values.

Figure 2: The overlap in dimension explanations aligns with the correlation between the node feature values for those dimensions. The dimension referenced by the blue subgraph shows a stronger correlation with the red dimensions and a lower correlation with the green dimension.

Figure 3: The node feature value indicates its proximity to the explanation substructure mapped to the corresponding dimension. The black node has a higher value for the dimension corresponding to the green subgraph (since it is 1 hop away) than for the dimension corresponding to the red subgraph (3 hops away).

et al., 2021; Dalmia et al., 2018) and heavily dependent on the specific embedding techniques used. Some approaches (Piaggesi et al., 2024) extend methods from the post-processing of word embeddings (Subramanian et al., 2018; Chen & Zaki, 2017) by minimizing reconstruction errors through over-complete auto-encoders to improve sparsity. Other works (Gogoglou et al., 2019; Khoshraftar et al., 2021; Dalmia et al., 2018) focus solely on extracting meaningful explanations, without addressing the underlying embedding learning process.

We propose DISENE (**Di**sentangled and **Se**lf-Explainable **N**ode **E**mbedding), a framework that addresses the interpretability gap in unsupervised node embeddings by generating inherently *self-explainable* node representations. In our approach, "self-explainable" means that each embedding dimension corresponds to a distinct subgraph that globally explains the structural information encoded within that dimension. These dimensional subgraphs highlight human-comprehensible functional components of the input graph, providing clear and meaningful insights. DISENE leverages *disentangled* representation learning, an approach that encodes latent variables corresponding to distinct factors of variation in the data (Wang et al., 2024), to produce node embeddings that are interpretable on a *per-dimension* basis.

In graph data, node behaviour is strongly influenced by mesoscale structures such as communities, which shape the network's organization and drive dynamics (Barrat et al., 2008). By leveraging disentangled representation learning, we capture these "topological" substructures more effectively, with each embedding dimension reflecting an independent graph unit (see Figure 1). We achieve this through a novel objective function that ensures *structural disentanglement*. Specifically, we optimize the embeddings so that each dimension is predictive of a unique substructure in the input graph. To avoid degenerate solutions, we incorporate an entropy-based regularizer that ensures the resulting substructures are non-empty and informative.

Our paradigm represents a shift in the language of explanations compared to the ones often considered when dealing with GNNs (Yuan et al., 2023). Explainability for GNNs often involves understanding which parts of the local computation graph (nodes, edges) and node attributes significantly influence the model's predictions (Funke et al., 2023; Ying et al., 2019; Schnake et al., 2022). On the other hand, the explanations that we aim to discover are inherently non-local, since they could involve mesoscale structures such as node clusters (Piaggesi et al., 2024), usually not included in the GNN computational graph.

We provide a comprehensive evaluation of the embeddings and uncover novel insights by proposing several new metrics that capture the interplay between disentanglement and explainability (see Section 3.3 for details). For instance, our *overlap consistency* metric (illustrated in Figure 2) shows that the overlap of the topological substructures used as explanations matches the correlation among the corresponding embedding dimensions, providing insights into the interdependencies among node characteristics within the graph. Moreover, we attribute meaning to the embedding values by showing that the magnitude of a node's entry in a specific dimension correlates with its proximity (as depicted in Figure 3) to the topological subgraphs associated

with that dimension. This relationship enhances our understanding of the relative positioning of nodes with respect to different graph components, thereby enhancing spatial awareness of the graph structure.

Contributions are summarized as follows: *(i)* we formalize *new and essential criteria* for achieving *disentangled and explainable node representations*, offering a fresh perspective on interpretability in unsupervised graph-based learning; *(ii)* we introduce *novel evaluation metrics* to help quantifying the goodness of node representation learning in disentangled and explainable settings *(iii)* we perform *extensive experimental analyses* on synthetic and real-world data to establish state-of-the-art results in self-explainable node feature learning. We release our code and data at `https://github.com/simonepiaggesi/disene`.

## 2 Preliminaries and Related Work

Given an undirected graph $\mathcal{G} = (\mathcal{V}, \mathcal{E})$, node embeddings are obtained through an encoding function $\mathbf{h} : \mathcal{V} \to \mathbb{R}^K$ that maps each node to a point in a $K-$dimensional vector space $\mathbb{R}^K$, where typically $D << |\mathcal{V}|$. We denote the $K$-dimensional embedding of a node $v \in \mathcal{V}$ as $\mathbf{h}(v) = [h_1(v), \dots, h_K(v)]$, where $h_d(v)$ represents the value of the $d$-th feature of the embedding for node $v$. Alternatively, we can represent all node embeddings collectively as a matrix $\mathbf{H}(\mathcal{G}) \in \mathbb{R}^{V \times K}$, where each entry $H_{vd} = h_d(v)$ corresponds to the $d$-th feature for node $v$. We can also refer to columns of such matrix, $\mathbf{H}_{:,d}$, as the dimensions of the embedding model space.

**Node embeddings interpretability.** Node embeddings are shallow encoding techniques, often based on matrix factorization or random walks (Qiu et al., 2018). Since the latent dimensions in these models are not aligned with high-level semantics (Senel et al., 2018; Prouteau et al., 2022), interpreting embeddings typically involves post-hoc explanations of their latent features (Gogoglou et al., 2019; Khoshraftar et al., 2021). Other works propose alternative methods to modify existing node embeddings, making them easier to explain with human-understandable graph features (Piaggesi et al., 2024; Shafi et al., 2024). From a different viewpoint, Shakespeare & Roth (2024) explore how understandable are the embedded distances between nodes. Similarly, Dalmia et al. (2018) investigate whether specific topological features are predictable, and then encoded, in node representations.

**Graph neural networks interpretability.** Graph Neural Networks (GNNs) (Wu et al., 2021) are deep models that operate via complex feature transformations and message passing. In recent years, GNNs have gained significant research attention, also in addressing the opaque decision-making process. Several approaches have been proposed to explain GNN decision process (Yuan et al., 2023), including perturbation approaches (Ying et al., 2019; Yuan et al., 2021; Funke et al., 2023), surrogate model-based methods (Vu & Thai, 2020; Huang et al., 2023), and gradients-based methods (Pope et al., 2019; Sánchez-Lengeling et al., 2020). In parallel, alternative research directions focused on concept-based explanations, i.e. high-level units of information that further facilitate human understandability (Magister et al., 2021; Xuanyuan et al., 2023).

**Disentangled learning on graphs.** Disentangled representation learning seeks to uncover and isolate the fundamental explanatory factors within data (Wang et al., 2024). In recent years, these techniques have gained traction for graph-structured data (Liu et al., 2020; Li et al., 2021; Yang et al., 2020; Fan & Gao, 2024). For instance, FactorGCN (Yang et al., 2020) disentangles an input graph into multiple factorized graphs, resulting in distinct disentangled feature spaces that are aggregated afterwards. IPGDN (Liu et al., 2020) proposes a disentanglement using a neighborhood routing mechanism, enforcing independence between the latent representations as a regularization term for GNN outputs. Meanwhile, DGCL (Li et al., 2021) focuses on learning disentangled graph-level representations through self-supervision, ensuring that the factorized components capture expressive information from distinct latent factors independently.

## 3 Our Proposed Framework: DiSeNE

In this section, we begin by outlining the key desiderata for achieving disentangled and self-explainable node representations. Next, we design a novel framework that meets these objectives by ensuring that the learned node representations are both disentangled and interpretable. Finally, we introduce new evaluation metrics to effectively assess the quality of node representation learning in both disentangled and explainable settings.

### 3.1 Core Objectives and Desiderata

In the context of unsupervised graph representation learning, we argue that learning self-explainable node embeddings amounts to reconstructing the input graph in a human-interpretable fashion. Traditionally, dot-product models based on NMF (Yang & Leskovec, 2013) and LPCA (Chanpuriya et al., 2023) decompose the set of graph nodes into clusters, where each entry of the node embedding vector represents the strength of the participation of the node to a cluster. In this scenario, the dot-product of node embeddings becomes intuitively understandable, as it reflects the extent of shared community memberships between nodes, thereby providing a clear interpretation of edge likelihoods. This concept is also related to distance encoding methods (Li et al., 2020; Klemmer et al., 2023), where a node feature $h_d(u)$ is expressed as a function of the node's proximity $\zeta(u, \mathcal{S}_d) = \mathrm{AGG}(\{\zeta(u,v), v \in \mathcal{S}_d\})$ to the *anchor set* $\mathcal{S}_d \subset \mathcal{V}$, using specific aggregation functions AGG. Typically, distance encodings are constructed by randomly sampling anchor sets (You et al., 2019), and used as augmented node features to enhance expressiveness and improve performance on downstream tasks.

Inspired by this idea, our goal is to optimize unsupervised node embeddings encoded by a GNN function $\mathbf{h} : \mathcal{V} \to \mathbb{R}^K$ trained on the graph $\mathcal{G} = (\mathcal{V}, \mathcal{E})$, such that node features resemble non-random, structurally meaningful anchor sets, thus improving human-interpretability. To achieve this, we propose three key desiderata for learning general-purpose node representations: *(i) connectivity preservation*, *(ii) dimensional interpretability*, and *(iii) structural disentanglement*. These desiderata serve as the foundational components of our approach, as detailed below.

**Connectivity preservation.** Ideally, node embeddings are constructed so that the geometric relationships in the low-dimensional space mirror the connectivity patterns of the original graph. Nodes with greater similarity in the network should be placed close to each other in the embedding space, and viceversa. To implement the approach, we train the node embedding function in recovering the graph structure. We employ a random walk optimization framework based on the skip-gram model with negative sampling (Huang et al., 2021). The loss function for this framework is defined as:

$$\mathcal{L}_{\mathrm{rw}} = - \sum_{(u,v) \sim P_{rw}} \log \sigma\big(\mathbf{h}(u)^\top \mathbf{h}(v)\big) - \sum_{(u',v) \sim P_n} \log \sigma\big(-\mathbf{h}(u')^\top \mathbf{h}(v)\big),$$

where $\sigma(\cdot)$ is the sigmoid function, $P_{rw}$ is the distribution of node pairs co-occurring on simulated random walks (positive samples), $P_n$ is a distribution over randomly sampled node pairs (negative samples), and $\mathbf{h}(u)^\top \mathbf{h}(v)$ represents the dot product between the embeddings of nodes $u$ and $v$. By optimizing this loss function, we encourage nodes that co-occur in random walks to have similar embeddings, effectively preserving the graph's structural information in the embedding space.

**Dimensional interpretability.** Embedding representations are multi-dimensional, with specific latent factors contributing in representing each dimension (e.g., social similarity, functional proximity, shared interests). A given edge between $u$ and $v$ may rely more heavily on certain dimensions. This perspective shows how local relationships (edges) are directly informed by the global structure of the embeddings, with each dimension contributing uniquely to reconstructing particular relationships. Given structure-preserving embeddings, meaning they effectively encode the input structure, we should be able to interpret each embedding dimension in terms of the graph's topological structure. Specifically, our framework attributes "responsibility" for reconstructing a local relationship (an edge) to specific dimensions, based on their contribution to the edge likelihood or the probability of reconstructing that edge through the embeddings. We achieve this by assigning local subgraphs to different latent dimensions. Consider the likelihood of an edge $(u, v)$, defined as $\hat{y}(u, v; \mathbf{h}) = \sigma\big( \sum_{d=1}^{K} h_d(u)h_d(v) \big)$. The likelihood score is obtained by applying the sigmoid to a linear combination of per-dimension products $h_d(u)h_d(v)$. Because the sigmoid is monotonic, each product's sign and magnitude directly determine its influence on $\hat{y}$. Thus, we can interpret the edge likelihood by inspecting the raw products. This decomposition requires only linear additivity and it does not depend on any statistical relationships among the terms. To understand how each dimension $d$ contributes to this likelihood, we compute the edge-wise dimension importance $\phi_d(u, v; \mathbf{h})$ as the deviation of the dimension-specific contribution

from its average over all edges:

$$\phi_d(u,v;\mathbf{h}) = h_d(u)h_d(v) - \frac{1}{|\mathcal{E}|}\sum_{(u',v')\in\mathcal{E}} h_d(u')h_d(v'). \tag{1}$$

Since the dot-product is a linear function $\sum_{d=1}^{K}\alpha_d h_d(u)h_d(v) + \beta$ with unitary coefficients $\alpha_d \equiv 1$ and zero intercept $\beta \equiv 0$, Eq. (1) corresponds to the formulation of LinearSHAP attribution scores (Lundberg & Lee, 2017), using the set of training edges as the background dataset. Essentially, the attribution function $\phi_d(u,v;\mathbf{h})$ indicates whether a specific dimension $d$ contributes positively to an edge's likelihood. A positive attribution score means that the dimension increases the likelihood of predicting the edge. If two dimensions are correlated, they may often push an edge with similar attributions, but each still provides an observable marginal deviation that we can measure. In other words, $\phi_d$ quantifies marginal responsibility, not exclusive responsibility; therefore it remains valid in the presence of correlations. Each dimension contributes to reconstructing specific edges or substructures, and together, these contributions create a unified representation of the entire graph. Thus, the global nature of embeddings emerges as a direct consequence of their contributions to local relationships. From this observation, we generate dimension-wise (global) explanations for the latent embedding model by collecting edge subsets with positive contributions:

$$\mathcal{E}_d = \{(u,v)\in\mathcal{E} : \phi_d(u,v;\mathbf{h}) > 0\}. \tag{2}$$

These self-explanations take the form of global edge masks $\mathbf{M}^{(d)} \in \mathbb{R}_{\geq 0}^{|\mathcal{V}|\times|\mathcal{V}|}$, where each entry is defined as $M_{uv}^{(d)}(\phi_d;\mathbf{h}) = \max\{0, \phi_d(u,v;\mathbf{h})\}$. By applying these masks to the adjacency matrix $\mathbf{A}$ through Hadamard product ($\odot$), we obtain $\mathbf{A}^{(d)} = \mathbf{A} \odot \mathbf{M}^{(d)}$. Each masked adjacency matrix $\mathbf{A}^{(d)}$ highlights the subgraph associated with dimension $d$. From these masked adjacency matrices, we construct edge-induced subgraphs $\mathcal{G}_d = (\mathcal{V}_d, \mathcal{E}_d)$, where $\mathcal{V}_d$ is the set of nodes involved in edges $\mathcal{E}_d$. These subgraphs act as anchor sets for the model, providing interpretable representations of how each embedding dimension relates to specific structural patterns within the graph. We will refer to edge-induced subgraphs computed as the aforementioned procedure (pseudo-code in Appendix E) as *explanation subgraphs/substructures* or *topological components* of the embedding model.

**Structural disentanglement.** To enhance the effectiveness of dimensionally interpretable encodings, each dimension of the latent space should encode an *independent* structure of the input graph, effectively acting as an anchor subgraph. Inspired by community-affiliation models (Yang & Leskovec, 2013; 2012), we introduce a node affiliation matrix $\mathbf{F} \in \mathbb{R}^{|\mathcal{V}|\times K}$ that captures the association between each node $u \in \mathcal{V}$ and anchor subgraph $\mathcal{G}_d = (\mathcal{V}_d, \mathcal{E}_d)$. Specifically, each entry $\mathbf{F}_{ud}$ is proportional to the magnitude of predicted meaningful connections between node $u$ and other nodes in $\mathcal{G}_d$, expressed using the per-dimension attribution scores from Eq. (1): $F_{ud}(\mathbf{h}) = \sum_{v\in\mathcal{V}_d}\phi_d(u,v;\mathbf{h})$. This aggregates the contributions of dimension $d$ to the likelihood of edges involving node $u$. To achieve structure-aware disentanglement, we enforce soft-orthogonality among the columns of the affiliation matrix[1]. This ensures that different embedding dimensions capture independent structures, leading to nearly non-overlapping sets of predicted links for each dimension. We express the columns of the affiliation matrix as $\mathbf{F}_{:,d}$ and obtain the disentanglement loss function as:

$$\mathcal{L}_{\text{dis}} = \sum_{d=1}^{K}\sum_{l=1}^{K}\left[\cos\left(\mathbf{F}_{:,d}, \mathbf{F}_{:,l}\right) - \delta_{d,l}\right] \tag{3}$$

where $\cos\left(\mathbf{F}_{:,d}, \mathbf{F}_{:,l}\right)$ denotes the cosine similarity between the $d$-th and the $l$-th columns of $\mathbf{F}$, and and $\delta_{d,l}$ is the Kronecker delta function (1 if $d = l$, 0 otherwise). This objective penalizes correlation between the edge importance attribution scores of different dimensions. By reducing redundancy, we ensure that each dimension contributes uniquely to the reconstruction of a small set of edges, promoting interpretability. This approach enables us to obtain disentangled representations (Wang et al., 2024), where embedding dimensions correspond to orthogonal latent factors of the input graph. Although higher-order disentanglement is possible (e.g., with groups of dimensions), we focus on single-feature disentanglement for achieving dimension-wise interpretability.

---

[1]Note that $\mathbf{F}_{ud}(\mathbf{h}) \equiv \sum_{v\in\mathcal{V}}h_d(u)h_d(v) - \frac{|\mathcal{V}|}{|\mathcal{E}|}\sum_{(u',v')\in\mathcal{E}}h_d(u')h_d(v')$ and assuming $|\mathcal{V}| << |\mathcal{E}|$, the second term becomes negligible, allowing us to approximate $\mathbf{F}_{ud}(\mathbf{h})$ and reduce computational costs.

### 3.2 Our Approach: DiSeNE

Building upon the above components, introduced to satisfy our desiderata for interpretable node embeddings, we present our approach, DiSeNE. Specifically, DiSeNE takes as input identity matrix $\mathbf{1}_{|\mathcal{V}| \times |\mathcal{V}|}$ as node attributes and, depending on the encoder architecture, also the adjacency matrix $\mathbf{A} \in \mathbb{R}^{|\mathcal{V}| \times |\mathcal{V}|}$. The input is encoded into an intermediate embedding layer $\mathbf{Z} \in \mathbb{R}^{|\mathcal{V}| \times D}$. Next, DiSeNE processes the embedding matrix $\mathbf{Z}$ to compute the likelihood of link formation between node pairs, given by $\hat{y}(u, v; \mathbf{h}) = \sigma(\mathbf{h}(u)^\top \mathbf{h}(v))$ where $\mathbf{h}(v) = \rho(\mathbf{W}^\top \mathbf{z}(v))$ are the final node representations in $\mathbf{H} \in \mathbb{R}^{|\mathcal{V}| \times K}$, obtained by applying a linear transformation $\mathbf{W} \in \mathbb{R}^{D \times K}$ followed by a non-linear activation function $\rho$. To encode $\mathbf{z}$, we employ architectures incorporating fully connected layers and graph convolutional layers (Wu et al., 2019). This process can be further enhanced by integrating more complex message-passing mechanisms or MLP operations. For example, the message-passing could initiate from an MLP-transformed node attribute matrix, MLP($\mathbf{X}$), or incorporate more sophisticated architectures beyond simple graph convolutions for increased expressiveness (Xu et al., 2019; Velickovic et al., 2017).

The embeddings are optimized by combining the previously described objective functions for preserving structural faithfulness and achieving structural disentanglement, thereby improving dimensional interpretability. To avoid degenerate disentanglement solutions we introduce a regularization strategy. Specifically, we aim avoiding the emergence of "empty" clusters characterized by near-zero columns in $\mathbf{F}$ that, while orthogonal to others, fail to convey meaningful information. This regularization ensures a minimal but significant level of connectivity within each topological substructure. Specifically, we enforce that the total amount of contributions to predicted edges in each anchor subgraph $\mathcal{G}_k$, $\sum_{u,v \in \mathcal{V}} \phi_k(u, v; \mathbf{h})$, to be non-zero. We found a more stable and precise approach by enforcing that the aggregated node features of each embedding dimension are non-zero, achieved by maximizing the entropy[2]: $\mathcal{H} = -\sum_{d=1}^{K} \left( \frac{\sum_u h_d(u)}{|| \sum_u \mathbf{h}(u) ||_1} \right) \log \left( \frac{\sum_u h_d(u)}{|| \sum_u \mathbf{h}(u) ||_1} \right)$. Thus, the model is optimized by minimizing the following comprehensive loss function:

$$\mathcal{L} = \mathcal{L}_{\text{rw}} + \mathcal{L}_{\text{dis}} + \lambda_{\text{ent}} \left( 1 - \frac{\mathcal{H}}{\log K} \right)$$

The hyperparameter $\lambda_{\text{ent}}$ determines the strength of the regularization, controlling the stability for explanation subgraph sizes across the various latent dimensions. We report the pseudo-code of DiSeNE in Appendix B, along with a space-time complexity analysis, showing that our method has $\mathcal{O}(|\mathcal{E}|K + |\mathcal{V}|K^2 + |\mathcal{V}|KTL)$ runtime complexity and $\mathcal{O}(|\mathcal{V}|K + |\mathcal{V}|L)$ space complexity ($T$ refers to the window size, $L$ to simulated walks length), which is in line with well-established techniques for node embeddings (Tsitsulin et al., 2021). Our approach deviates from typical GNNs by focusing solely on learning from graph topology, as node attributes may not always align with structural information (e.g., in cases of heterophily (Zhu et al., 2024)). While semantic features can be integrated in various ways (Tan et al., 2024), we chose a more straightforward, broadly applicable method. Our node embeddings produce interpretable structural features $\mathbf{h}(u)$, which can be concatenated with node semantic attributes $\mathbf{x}(u)$, enabling transparent feature sets for effective explanations in downstream tasks via post-hoc tabular techniques.

### 3.3 Proposed Evaluation Metrics

In the following, we present novel metrics to quantify interpretability and disentanglement in unsupervised node embeddings, which we use to compare models in our experiments. Unlike prior works, which primarily focus on explaining graph model decisions, our approach offers a novel perspective by targeting the explanation of graph model encodings. Traditional graph-based explainers focus on interpreting GNN model decisions for specific tasks, highlighting subgraphs or motifs responsible for the predictions (Longa et al., 2025; Li et al., 2025). In contrast, our objective is to assess the interpretability of GNN models in a task-agnostic setting, by evaluating explanatory subgraphs derived from node embedding dimensions that are learned without supervision. In the next, we define our comprehensive vocabulary of evaluation metrics. While alternative taxonomies exist in the literature, we propose a consistent framework highlighting similarities and differences

---

[2]We assume non-negative activation functions $\rho$ (e.g., ReLU), thus every post-activation feature $h_d(u)$ is itself non-negative. Therefore, their sum $\sum_d h_d(u)$ is non-negative as well, and an absolute-value operator in the entropy mass term is unnecessary.

between our indicators and other established metrics in the field of graph-based explainability (Yuan et al., 2023; Kakkad et al., 2023).

**Topological Alignment.** This metric measures how well the explanations of embedding dimensions align with human-interpretable graph structures. Given their importance in the organization of complex real-world systems (Girvan & Newman, 2002; Hric et al., 2014), community modules (or clusters) are often the most intuitive units for understanding a graph. We evaluate topological alignment by handling edges in explanation masks $\{\mathbf{M}^{(d)}\}_{d=1,\dots,K}$ as retrieved items from a query, and measuring their overlap with the edges in the ground-truth communities using precision, recall, and $F_1$-score. While this metric captures one aspect of human-comprehensibility, it does not exhaust the space of interpretable structures, such as motifs, roles, or domain-specific patterns. For example, in biological networks, structures like protein complexes or regulatory circuits are often considered as more interpretable. In social networks, roles like hubs, bridges, or peripheral nodes may give better explanations than just communities. Future work could incorporate alternative structural annotations to evaluate broader forms of topological alignment.

Let $\mathcal{C}(\mathcal{E}) = \{\mathcal{C}^{(1)}, \dots, \mathcal{C}^{(m)}\}$ denotes the set of truthful link communities of the input graph[3]. Associated to partition $\mathcal{C}^{(i)}$, we define ground-truth edge masks $\mathbf{C}^{(i)} \in \{0,1\}^{V \times V}$ with binary entries $C_{uv}^{(i)} = \mathbb{1}[(u,v) \in \mathcal{C}^{(i)}]$. For a given mask $\mathbf{M}^{(d)}$, *topological alignment* score is given by the maximum edge overlap (as $F_1$-score) of the explanation substructure computed across community structures in $\mathcal{C}(\mathcal{E})$, quantifying how closely the explanation match human-understandable ground-truth:

$$\text{Align}(\mathbf{M}^{(d)}) = \max_i \left\{ F_1(\mathbf{M}^{(d)}, \mathbf{C}^{(i)}) \right\} = \max_i \left\{ \frac{2}{\text{prec}(\mathbf{M}^{(d)}, \mathbf{C}^{(i)})^{-1} + \text{rec}(\mathbf{M}^{(d)}, \mathbf{C}^{(i)})^{-1}} \right\} \tag{4}$$

For precision, we weigh relevant item scores with normalized embedding masks values: $\text{prec}(\mathbf{M}^{(d)}, \mathbf{C}^{(i)}) = \frac{\sum_{u,v} M_{uv}^{(d)} C_{uv}^{(i)}}{\sum_{u,v} M_{uv}^{(d)}}$. For recall, we weigh binarized embedding masks values with normalized ground-truth scores[4]: $\text{rec}(\mathbf{M}^{(d)}, \mathbf{C}^{(i)}) = \frac{\sum_{u,v} \mathbb{1}[M_{uv}^{(d)} > 0] C_{uv}^{(i)}}{|\mathcal{C}^{(i)}|}$. This approach is similar to the *accuracy* assessment used in GNNExplainer and subsequent works (Ying et al., 2019; Luo et al., 2020), where explanations for model decisions are compared to planted graph substructures, perceived as human-readable justifications for a node's prediction. However, instead of evaluating individual decisions' explanations by using synthetic ground-truth motifs, we perform a global assessment of the unsupervised embedding dimensions by comparing their explanation subgraphs with a reasonable ground-truth about the graph structure. An analogous evaluation of explanation correctness which specifically focuses on individual model decisions will be provided later by the *plausibility* metric.

**Sparsity.** We refer to sparsity as a measure of the localization of subgraph explanations, it is generally defined as the ratio of the number of bits needed to encode an explanation compared to those required to encode the input (Pope et al., 2019; Funke et al., 2023). As we produce soft masks, we use entropy for this quantification. Given that compact explanations are more effective in delivering clear insights, we evaluate *sparsity* by measuring the normalized Shannon entropy over the mask distribution:

$$\text{Sp}(\mathbf{M}^{(d)}) = -\frac{1}{\log |\mathcal{E}|} \sum_{(u,v) \in \mathcal{E}} \left( \frac{M_{uv}^{(d)}}{\sum_{u',v'} M_{u'v'}^{(d)}} \right) \log \left( \frac{M_{uv}^{(d)}}{\sum_{u',v'} M_{u'v'}^{(d)}} \right). \tag{5}$$

A lower entropy in the mask distribution indicates higher sparsity/compactness. Motivation for using entropy for explanation size quantification can be found in one of the existing works (Funke et al., 2023).

**Overlap Consistency.** This metric assesses whether the correlation between two embedding dimensions is mirrored in their corresponding explanations. A well-structured, disentangled latent space should correspond

---

[3]Synthetic graphs can be constructed with ground-truth relevant sub-structures (like BA-Shapes (Ying et al., 2019) or SBM graphs). In real-world graphs, it is usually reasonable to assume that the community structure (Fortunato, 2010) can serve as ground-truth.

[4]For the precision, we normalize with the sum of scores because they are continuous. For recall, we use the cardinality in place of the sum because the ground-truth has binary scores.

to distinct, uncorrelated topological structures. For example, if two embedding dimensions are correlated, their explanation substructures should also overlap. In our context, this a proxy measure for explanation's *contrastivity* (Wang et al., 2023; Pope et al., 2019), based on the intuition that explanations for unrelated dimensions should differ substantially—particularly in the specific substructures they highlight. We aim to quantify how different topological components affect pairwise feature correlations in the latent space. To achieve this, we propose a metric that measures the strength of association between the physical overlap of the explanation substructures $\{\mathcal{G}_d\}$ and the correlation among corresponding latent dimensions $\{\mathbf{H}_{:,d}\}$. We compute the overlap between two subgraph components using the Jaccard similarity index of their edge sets from Eq. (2): $J(d,l;\mathbf{h}) = \frac{|\mathcal{E}_d \cap \mathcal{E}_l|}{|\mathcal{E}d \cup \mathcal{E}_l|}$. The *overlap consistency* (OvC) metric measures the linear correlation between pairwise Jaccard values and squared Pearson correlation coefficients ($\rho^2$) of the embedding features:

$$\text{OvC}(\mathbf{h}) = \rho\Big([J(d,l;\mathbf{h})]_{d<l}, [\rho^2(\mathbf{H}_{:,d}, \mathbf{H}_{:,l})]_{d<l}\Big) \tag{6}$$

where $[*]_{d<l}$ denotes the condensed list of pair-wise similarities. By using $\rho^2$ we remain agnostic about the sign of the correlation among latent features, since high overlaps could originate from both cases.

**Positional Coherence.** This metric evaluates whether the feature value of a node representation in a specific embedding dimension corresponds to its spatial relationship with the explanation substructure for that dimension. In fact, an effective representation should preserve meaningful spatial properties that reflect node proximity and connectivity patterns. In our context, this is a proxy measure for explanation's *faithfulness* (Zhao et al., 2023b), reflecting how well the dimension-based structures align with the embeddings they are intended to explain. To achieve this, we propose to measure the extent to which node entries reflect their relative positions with the subgraphs used as explanations. For example, if node $u$ has a high value in embedding dimension $d$, it should be positioned closer to the substructure that explains dimension $d$. Typically, positional encoding (Rampásek et al., 2022; Li et al., 2020; You et al., 2019) involves the use of several sets of node *anchors* $\mathcal{S}_d \subset \mathcal{V}$ that establish an intrinsic coordinate system. This system influences the node $u$'s features based on the node's proximity $\zeta(u, \mathcal{S}_d) = \text{AGG}(\{\zeta(u,v), v \in \mathcal{S}_d\})$, where AGG denotes a specific pooling operation. As node proximity, we used the inverse of the shortest path distance $\zeta_{spd}(u,v) \equiv (1 + d_{spd}(u,v))^{-1}$. As the anchor sets, we chose the embedding substructures used for explanations, $\mathcal{S}_d \equiv \mathcal{V}_d$. For a specified pair of dimensions $(d,l)$, we assess the correlation between node features along dimension $d$ and the corresponding distances to the topological component indexed by $l$ via feature-proximity correlation: $fp_{corr}(d,l;\mathbf{h}) = \rho\Big([\zeta_{spd}(u, \mathcal{V}_d)]_{u\in\mathcal{V}}, \mathbf{H}_{:,l}\Big)$. The *positional coherence* metric (PoC) is defined to specifically evaluate the degree to which each feature $d$ is uniquely correlated with its corresponding topological component $\mathcal{V}_d$, without being significantly influenced by correlations with other substructures. This metric is calculated as the ratio of the average $fp_{corr}$ for the given dimensions to the average $fp_{corr}$ computed with pairs of permuted dimensions:

$$\text{PoC}(\mathbf{h}) = \frac{\sum_d fp_{corr}(d,d;\mathbf{h})}{\left\langle \sum_d fp_{corr}(d,\pi(d);\mathbf{h}) \right\rangle_\pi} \tag{7}$$

where $\langle . \rangle_\pi$ denotes an empirical average over multiple permutations. By comparing with random feature-subgraph pairs, the metric avoid promoting models with redundancies in the latent features, where high correlations with other topological components are possible.

**Plausibility.** Plausibility evaluates how closely subgraph explanations align with human reasoning, a concept also studied in existing works like BAGEL (Rathee et al., 2022). Unlike topological alignment, which measures the alignment of explanation substructures with community clusters in the graph (independent of task-specific information), plausibility focuses on explanations for decisions made in downstream tasks, using the embedding features as predictor variables. To this end, relying on feature-based post-hoc explanation methods (Lundberg & Lee, 2017; Ribeiro et al., 2016), we construct instance-specific explanations to determine feature importance scores related to the topological structures. Typical feature importance explainers (Bodria et al., 2023) are useless in this context because node embeddings have inherently uninterpretable features, leading to uninformative explanations. Our approach overcomes this limitation by mapping explanations back to the graph's structural components, that are human-readable.

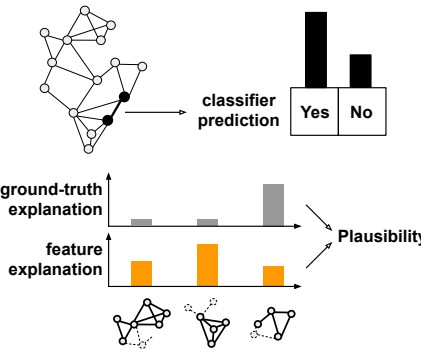

We detail the procedure for link prediction here (see also Figure 4), but we also report the methodology for node classification in Appendix E. The process involves three key steps.

*(1) Training a downstream classifier.* We train a binary classifier $b : \mathbb{R}^K \to [0,1]$ to perform the downstream link prediction task using node embeddings. This classification model can either leverage feature interpretability techniques, such as SHAP (Lundberg & Lee, 2017), or being inherently interpretable, such as logistic regression.

*(2) Extracting post-hoc explanations.* For an edge instance $\mathbf{h}(u,v)$ (which could be derived from node-pair operations such as $\mathbf{h}(u) \odot \mathbf{h}(v)$), we employ post-hoc methods to determine the feature relevance for the classifier prediction on the node pair instance $(u,v)$, $\{\Psi_j(u,v;b)\}_{j=1,\ldots,K}$ for each of the embedding dimensions. Similarly to Eq. (1) which defines task-agnostic masks, here we calculate task-specific masks $\mathbf{B}^{(j)} \in \mathbb{R}^{V \times V}_{\geq 0}$ which aggregate the logic of the classifier

Figure 4: Sketch of Plausibility metric computation. High plausibility scores indicate that the dimensions deemed more comprehensible also received higher importance scores from the post-hoc attribution technique.

based on individual feature importance: $B_{uv}^{(j)}(\Psi_j;b) = \max\{0, \Psi_j(u,v;b)\}$. Substructures corresponding to the most important features (identified by the post-hoc techniques), or the explanations directly returned by the interpretable model serve as the final explanations.

*(3) Evaluation.* Explanations are then compared with the available human rationale behind the decisions. To do that, we resort the $F_1$-score in Eq. (4) with respect to the ground-truth structure of the edge under study, indexed by $g(u,v)$, that it is known has driven the classifier decision. Specifically, we define *plausibility* for an individual prediction $b(u,v)$ as the average $F_1$-accuracy relative to the ground-truth structure for the instance, weighted by the computed feature importance:

$$Pl(u,v;b) = \frac{\sum_{j=1}^{K} f(\Psi_j(u,v;b)) F_1(\mathbf{B}^{(j)}, \mathbf{C}^{g(u,v)})}{\sum_{j=1}^{K} f(\Psi_j(u,v;b))} \tag{8}$$

where $f$ is a function guaranteeing the non-negativity of relevance weights. This ensures that only the features that are both interpretable and significant to the local prediction contribute substantially to the score, penalizing instead those important features that are not human-comprehensible.

It is worth to note that although our measure is similar to the accuracy index used to assess correctness of explanation subgraphs (Ying et al., 2019), we refer to it as "plausibility." This is because, despite their competitive classification performance, typical embedding representations may fail to capture the truthful substructures in the most task-predictive features, as they are not constrained to follow human reasoning. Thus, with plausibility, we aim to understand when predictive embedding features are also congruent to human rationales. The choice of the weighting function $f$ is case-specific and tailored to the objective of the analysis. For example, choosing $f$ as indicator function such that $f(\Psi_j) = \mathbb{1}[j = \text{argmax}\{\Psi_1, \ldots \Psi_K\}]$ allows to study the plausibility of the most predictive substructure.

## 4 Experiments

We conduct extensive experiments to answer the following research questions:

**(RQ1) Human understandability:** How comprehensible and sparse are the explanation substructures generated by DISENE?
**(RQ2) Structural disentanglement:** Do the disentangled subgraphs reveal intrinsic properties of node embeddings, like feature correlations and latent positions?
**(RQ3) Utility for downstream tasks:** Are the identified substructures plausible and coherent enough to serve as explanations in downstream tasks?

To address our research questions, we extract topological components from multiple embedding methods, trained on different graph data, by computing edge subsets defined in Eq. (2), and analyzing embedding metrics defined in Section 3.3. In the following, we describe the data, models, experimental setup, and results.

Moreover, in the Appendix we report the following supplementary experiments: in section D, we analyze accuracy-interpretability trade-offs for different embedding methods; in section G, we perform ablation studies on the interpretability performances of DiSeNE while varying the entropy regularization and the depth of the convolutional encoder; in section H, we showcase the human-understandability of embeddings in the presence of rich topic and text information in graph data.

### 4.1 Datasets and Competitors

**Datasets.** We ran experiments on four real-world datasets (Cora, Wiki, FaceBook, PPI), and six synthetic datasets (Ring-of-Cliques, SBM, BA-Cliques, ER-Cliques, Tree-Cliques and Tree-Grids) with planted subgraphs, which serve as ground-truth human rationales for explaining specific node labels. Statistics for these datasets are provided in Table A1 in the Appendix. Additionally we employ several biological datasets (see Appendix C) for the evaluation on multi-label node classification. BA-Cliques and ER-Cliques are variations of the BA-Shapes (Ying et al., 2019) where we randomly attach cliques, instead of house motifs, to Barabási-Albert and Erdős-Rényi random graphs. Tree-Cliques and Tree-Grids (Ying et al., 2019) are composed of a 8-level balanced tree, with cliques and 3x3 grid motifs respectively. Ring-Cliques and SBM (Abbe, 2017) are implemented in NetworkX[5]. For synthetic data, we present only results for plausibility metrics, leaving the other findings in the Appendix F.

**Methods.** We compare with different node embedding methods in producing explainable feature dimensions. Competitors include shallow encoders DeepWalk (Perozzi et al., 2014), InfWalk (Chanpuriya & Musco, 2020), and deep graph models Graph Autoencoder (GraphAE) (Salha et al., 2020), GraphSAGE (Hamilton et al., 2017) and DGLFRM (Mehta et al., 2019). We also apply the Dine retrofitting approach (Piaggesi et al., 2024) to post-process embeddings from DeepWalk and GraphAE. Moreover, we compare with graph-based local explanation methods GNNExplainer (Ying et al., 2019) and PGExplainer (Luo et al., 2020). Post-hoc explainers are applied to the output of different GNN models for node classification: GCN (Wu et al., 2019), GraphSAGE (Hamilton et al., 2017), and GATv2 (Brody et al., 2022). We evaluate our method DiSeNE in two variants: a 1-layer fully-connected encoder (DiSe-FCAE) and a 1-layer convolutional encoder (DiSe-GAE). GNN-based methods are trained using the identity matrix as node features. Specific training settings are provided in Appendix A.

**Setup.** In experiments on real-world graphs, we investigate latent space interpretability and disentanglement metrics by keeping the output embedding dimension fixed at 128. This dimensionality was chosen to ensure that all methods achieve optimal performance in terms of test accuracy, specifically for link prediction (see the Appendix sections C and D for extensive results on downstream tasks and embedding metrics). For synthetic data, since we investigated plausibility metric referred to a downstream classifier, thus we did not focus on a specific dimension but we selected the best score metric varying the output dimensions in the list $[2, 4, 8, 16, 32, 64, 128]$. Each reported result is an average over 5 runs. For link prediction, we use a 90%/10% train/test split, and for node classification, we use an 80%/20% split. All results refer to the training set, except for downstream task experiments, where we present results for the test set.

### 4.2 Results and Discussion

**(RQ1) Are the topological substructures both comprehensible and sparse to support human understandability?** Here we explore how well the represented topological structures can serve as global explanations for node embeddings, quantifying the Topological alignment in the terms of associations between model parameters and human-understandable units of the input graph, as well as the Sparsity of these associations. For Topological alignment, we apply modularity maximization to find meaningful clusters (Blondel et al., 2023). In Table 1 we show compact scores as the average values $\frac{1}{K} \sum_{d=1}^{K} \mathrm{Align}(\mathbf{M}_d)$ and $1 - \frac{1}{K} \sum_{d=1}^{K} \mathrm{Sp}(\mathbf{M}_d)$ over all the embedding features. Since for sparsity we report the value subtracted from 1, all the scores present better results with higher values.

DeepWalk, InfWalk and GraphSAGE show moderate performance in **Topological Alignment**, excelling slightly on FB but underperforming on PPI, while GraphAE consistently lags behind, particularly on Wiki

---

[5]https://networkx.org/documentation/stable/reference/generators.html

Table 1: Topological alignment and sparsity in real data. Best scores in bold, second best underlined.

| Method | Topological Alignment | | | | Sparsity | | | |
|---|---|---|---|---|---|---|---|---|
| | CORA | WIKI | FB | PPI | CORA | WIKI | FB | PPI |
| DEEPWALK | .363±.003 | .356±.002 | .602±.004 | .281±.002 | .183±.001 | .165±.002 | .130±.004 | .136±.003 |
| GRAPHAE | .299±.001 | .248±.002 | .481±.014 | .263±.003 | .182±.002 | .164±.004 | .154±.003 | .135±.003 |
| INFWALK | .281±.002 | .288±.001 | .658±.006 | .312±.002 | .211±.003 | .185±.002 | .318±.010 | .177±.002 |
| GRAPHSAGE | .358±.004 | .307±.007 | .583±.003 | .306±.005 | .189±.001 | .189±.003 | .145±.002 | .172±.001 |
| DGLFRM | .551±.015 | .555±.014 | .618±.005 | .515±.008 | .314±.014 | .346±.024 | **.373**±.018 | **.383**±.012 |
| DW+DINE | .511±.051 | .496±.014 | .813±.025 | **.569**±.022 | .317±.036 | .266±.007 | .226±.009 | .188±.002 |
| GAE+DINE | .569±.004 | .591±.004 | .843±.005 | .484±.007 | .290±.001 | .252±.001 | .195±.002 | .198±.002 |
| DISE-FCAE | .822±.001 | .755±.003 | **.971**±.001 | .484±.001 | **.504**±.001 | **.419**±.001 | .297±.002 | .282±.001 |
| DISE-GAE | **.834**±.003 | **.762**±.004 | .967±.001 | .515±.001 | .496±.001 | .418±.001 | .304±.003 | .254±.002 |

Table 2: Overlap consistency and positional coherence in real data. Best scores in bold, second best underlined.

| Method | Overlap Consistency | | | | Positional Coherence | | | |
|---|---|---|---|---|---|---|---|---|
| | CORA | WIKI | FB | PPI | CORA | WIKI | FB | PPI |
| DEEPWALK | .137±.009 | .143±.006 | .115±.007 | .015±.003 | 1.078±0.025 | 0.835±0.025 | 1.119±0.015 | 1.009±0.015 |
| GRAPHAE | .269±.002 | .295±.004 | .273±.017 | .452±.008 | 1.023±0.006 | 1.040±0.002 | 1.001±0.013 | 1.016±0.001 |
| INFWALK | .008±.003 | .023±.002 | .021±.002 | .134±.002 | 1.004±0.011 | 0.998±0.004 | 0.938±0.053 | 0.999±0.002 |
| GRAPHSAGE | .211±.003 | .136±.017 | .230±.007 | .097±.040 | 1.099±0.012 | 1.103±0.010 | 1.005±0.007 | 1.018±0.002 |
| DGLFRM | .061±.004 | .047±.008 | .070±.006 | .102±.002 | 1.263±0.052 | 1.197±0.043 | 1.349±0.065 | **1.291**±0.049 |
| DW+DINE | .900±.012 | .804±.032 | .851±.017 | .855±.016 | 1.790±0.076 | 2.126±0.065 | 1.792±0.058 | 1.247±0.043 |
| GAE+DINE | .560±.010 | .610±.006 | .801±.016 | .646±.003 | 2.317±0.028 | 2.551±0.048 | 1.783±0.037 | 1.098±0.004 |
| DISE-FCAE | .956±.001 | .941±.004 | **.963**±.004 | .934±.003 | 5.210±0.080 | 3.540±0.082 | 3.348±0.085 | 1.283±0.004 |
| DISE-GAE | **.961**±.001 | **.944**±.005 | .954±.002 | **.937**±.001 | **5.300**±0.193 | **4.343**±0.144 | **3.388**±0.054 | 1.261±0.005 |

and PPI. DGLFRM shows good topological alignment across all datasets. Incorporating DINE improves results, especially for GAE+DINE, which achieves improved scores on all datasets. The proposed models, DISE-FCAE and DISE-GAE, deliver the highest overall performance. DISE-FCAE performs well on FB, while DISE-GAE excels across CORA and WIKI. However, both models show sub-optimal results on PPI, suggesting potential for further improvement on this dataset.

DEEPWALK and GRAPHAE offer moderate **Sparsity**, peaking on CORA, but underperform on other datasets. INFWALK excels on FB but shows moderate results elsewhere, while GRAPHSAGE performs poorly in terms of sparsity across all datasets. DEEPWALK and GAE significantly improve their sparsity with DINE, particularly on CORA. DGLFRM shows competitive sparsity, excelling in FB and PPI. For the proposed models, DISE-FCAE and DISE-GAE perform best across datasets CORA, WIKI.

**(RQ2) Can the identified subgraphs explain the intrinsic characteristics of the node embeddings?**
Here we explore how well the defined topological units represent information in the node embedding space, providing insights into how the relative and absolute positioning of topological structures influences the feature encoding within a graph. By quantifying these relationships, we can better understand the underlying patterns and structural information encoded in graph embeddings. In Table 2 we report Positional Coherence and Overlap Consistency for the examined embedding methods. For the second metric, as node proximity we used the inverse of the shortest path distance with `sum` as pooling.

DEEPWALK, INFWALK and DGLFRM perform poorly for **Overlap Consistency**, while GRAPHAE shows moderate scores, particularly on PPI. GRAPHSAGE performs slightly worse, with the best overlap consistency on FB and CORA. DW+DINE achieves strong scores across all datasets, while GAE+DINE performs solidly but slightly lower, with its best result on FB. The proposed models, DISE-FCAE and DISE-GAE, outperform all others, achieving the highest consistency across all datasets except on CORA. DISE-FCAE excels on FB and WIKI, while DISE-GAE achieves the best overall score on PPI.

DEEPWALK, GRAPHAE, and GRAPHSAGE demonstrate moderate **Positional Coherence**. INFWALK consistently scores around 1.0 on all datasets, indicating stable but unremarkable coherence. Incorporating DINE leads to substantial improvements for both DEEPWALK and GAE, achieving notable gains on CORA,

Table 3: Plausibility for node embeddings in synthetic data. Best scores in bold, second best underlined.

| Method | Link Prediction | | | | Node Classification | | | |
|---|---|---|---|---|---|---|---|---|
| | Ring-Cl | SBM | BA-Cl | ER-Cl | BA-Cl | ER-Cl | Tr-Cl | Tr-Gr |
| DeepWalk | .234±.003 | .205±.008 | .173±.002 | .160±.006 | .146±.002 | .141±.003 | .103±.007 | .091±.002 |
| GraphAE | .183±.003 | .160±.002 | .145±.004 | .145±.005 | .130±.002 | .135±.006 | .083±.001 | .072±.001 |
| InfWalk | .224±.005 | .181±.005 | .218±.007 | .212±.008 | .129±.002 | .141±.004 | .097±.002 | .093±.004 |
| GraphSAGE | .252±.005 | .217±.003 | .186±.006 | .178±.005 | .160±.004 | .154±.002 | .093±.002 | .084±.003 |
| DGLFRM | .343±.006 | .224±.002 | .220±.010 | .210±.007 | .149±.003 | .143±.002 | .166±.001 | .175±.009 |
| DW+Dine | .943±.012 | .904±.002 | .744±.008 | .724±.040 | .320±.031 | .327±.008 | .549±.015 | .627±.004 |
| GAE+Dine | .549±.005 | .547±.014 | .418±.011 | .387±.002 | .351±.011 | .397±.003 | .366±.013 | .254±.005 |
| DiSe-FCAE | **.978**±.001 | **.924**±.006 | **.950**±.006 | .938±.014 | **.820**±.011 | .791±.012 | **.860**±.004 | **.810**±.008 |
| DiSe-GAE | .969±.002 | .910±.006 | .936±.003 | **.941**±.005 | .813±.003 | **.797**±.009 | .791±.005 | .800±.004 |

Table 4: Plausibility for graph explainers in synthetic data. Best scores in bold, second best underlined.

| Node Classif. Dataset | GnnExplainer | | | PGExplainer | | | DiSeNE | |
|---|---|---|---|---|---|---|---|---|
| | GCN | GSAGE | GATv2 | GCN | GSAGE | GATv2 | FCAE | GAE |
| BA-Cl | .729±.004 | .703±.006 | .707±.004 | .895±.005 | .581±.038 | .596±.016 | **.919**±.001 | .875±.009 |
| ER-Cl | .638±.005 | .611±.005 | .633±.002 | **.923**±.004 | .704±.032 | .724±.002 | .881±.006 | .872±.008 |
| Tr-Cl | .846±.004 | .829±.005 | .832±.006 | .863±.009 | .374±.006 | .422±.074 | **.926**±.001 | .871±.005 |
| Tr-Gr | .847±.005 | .833±.004 | .832±.003 | .573±.003 | .641±.045 | .765±.039 | .889±.006 | **.898**±.001 |

Wiki and FB. DGLFRM shows moderate scores, nevertheless scoring the best in PPI. The proposed models, DiSe-FCAE and DiSe-GAE, far outperform other methods, with DiSe-FCAE achieving top scores on PPI, like DGLFRM, while DiSe-GAE dominates on Cora, Wiki and FB (though with higher variance): both models show consistent superiority.

**(RQ3) Are the identified latent structures sufficiently meaningful to serve as explanations for downstream tasks?** Node embeddings serve as versatile feature representations suitable for downstream tasks, though they typically function as "tabular-like" feature vectors without semantic labels for each feature. This limitation restricts the use of established post-hoc analysis methods (Bodria et al., 2023) like LIME, SHAP, etc. Our method allows us to link topological substructures with embedding features, thereby assigning semantic labels to node vectors. Consequently, we are able to explain a downstream classifier trained with unsupervised embeddings using feature attribution. Our goal is to assess whether the task-important features align with human understanding by measuring the Plausibility.

In these experiments we consider node classification and link prediction as binary downstream tasks, training a logistic regression classifier $b(x; \boldsymbol{\beta}) = \sigma(\sum_{j=1}^{K} \beta_j h_j(x) + \beta_0)$, where $x$ is a node/link instance. We use SHAP (Lundberg & Lee, 2017) to compute the instance-wise feature attribution values $\{\Psi_j(x; b)\}_{j=1...K}$. For node classification, we consider positive instances as the nodes inside a clique in the synthetic graph. Accordingly, the ground-truth explanation for a node is the set of nodes within the clique it belongs to. For link prediction, we focus on test edges that were inside a clique before removal, where the ground-truth explanation is again the set of edges inside the clique itself. We compute plausibility scores over test instances with correct predicted label, because local explanations extracted from wrong predictions are not reliable for analyzing model decisions. We report in Appendix E the corresponding accuracy scores of downstream classifiers.

Table 3 compares **Plausibility** scores where node features from different embeddings are used to train downstream predictors. Since typical embeddings exhibit semantic patterns distributed across many dimensions (Elhage et al., 2022), here we consider all the contributing dimensions employing a non-negative weighting function $f(*) = \max(0, *)$. This choice prevents bias toward methods that inherently produce disentangled semantics (e.g., by analyzing only top-ranked dimensions). DeepWalk, GraphAE, and InfWalk perform modestly, with DeepWalk scoring the highest among these on Ring-Cl and InfWalk showing relative strength on BA-Cl. Compared with earlier approaches, DGLFRM delivers a modest gain in Plausibility, most notably in link prediction, whereas GraphSAGE significantly underperforms across all tasks, especially in node classification. The addition of Dine improves both DeepWalk and GAE. DW+Dine excels with strong performance on Ring-Cl, SBM, and Tree datasets, while GAE+Dine achieves slightly worst results,

particularly on node classification tasks, such as in TR-GR. Within the proposed models, DISE-FCAE and DISE-GAE consistently achieve the highest scores ranking as the best two methods overall.

In Table 4 we compare **Plausibility** for state-of-the-art local post-hoc explainers for graphs in node classification. We emphasize that our approach focuses on explaining model encodings, unlike methods such as GNNEXPLAINER and PGEXPLAINER, which explain model decisions. These methods present local explanation in the form of node and/or edge importance, whereas in our method, combined with feature-based explainer, the explanation format is a vector of feature importance, associated with a subgraph for each feature. To make a suitable comparison, we consider as the explanation presented by our method the subgraph associated to the most important embedding feature (according to the logistic classifier). Recalling Eq. (8), this approach is equivalent to choosing the function $f(\Psi_j) = \mathbb{K}[j = \mathrm{argmax}\{\Psi_1, \ldots \Psi_K\}]$ to compute plausibility index. We observe GNNEXPLAINER has uniform results across different input GNN models, instead PGEXPLAINER performs best with GCN. DISE-FCAE and DISE-GAE outperforms the competitors in most of the cases, except with GCN+PGEXPLAINER in ER-CLIQUES. Reported results show that our method is capable of producing subgraph-based local explanations with comparable, or even better, plausibility scores than GNNEXPLAINER/PGEXPLAINER. To enhance clarity, Appendix E includes qualitative examples that visualize the local explanations generated by the different methods.

## 5 Conclusions

We present DISENE, a novel framework for generating self-explainable unsupervised node embeddings. To build our framework, we design new objective functions that ensure connectivity preservation, dimensional explainability, and structural disentanglement. Unlike traditional GNN explanation methods that typically extract a subgraph from a node's local neighborhood, DISENE introduces a paradigm shift by learning node embeddings where each dimension captures an independent structural feature of the input graph. Additionally, we propose new metrics to evaluate the human interpretability of explanations, analyze the influence of spatial structures and node positions on latent features, and apply post-hoc feature attribution methods to derive task-specific instance-wise explanations. Our results show that interpretable node representations for graphs can be obtained by disentangling topological substructures across embedding dimensions. Additionally, the most important node features identified by post-hoc techniques aligns with the true explanation subgraphs. These findings mark a significant step toward human-centric evaluations of node embeddings, pointing to critical directions for future work in advancing human-in-the-loop validations in graph feature learning.

## Acknowledgments

S. Piaggesi acknowledges support from the European Community program under the funding schemes: ERC-2018-ADG G.A. 834756 "XAI: Science and technology for the eXplanation of AI decision making".

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

# Appendix

Table A1: Summary statistics of graph-structured data. In empirical data, we restrict our analysis to the largest connected component of any graph.

|  | Cora | Wiki | FB | PPI | Ring-Cl | SBM | BA-Cl | ER-Cl | Tr-Cl | Tr-Gr |
|---|---|---|---|---|---|---|---|---|---|---|
| # nodes | 2,485 | 2,357 | 4,039 | 3,480 | 320 | 320 | 640 | 640 | 831 | 799 |
| # edges | 5,069 | 11,592 | 88,234 | 53,377 | 1,619 | 1,957 | 3,138 | 4,196 | 2,081 | 972 |
| # clusters/motifs | 28 | 18 | 16 | 9 | 32 | 32 | 32 | 32 | 32 | 32 |
| density | 0.002 | 0.004 | 0.011 | 0.009 | 0.032 | 0.038 | 0.015 | 0.021 | 0.006 | 0.003 |
| clust. coeff. | 0.238 | 0.383 | 0.606 | 0.173 | 0.807 | 0.561 | 0.486 | 0.456 | 0.360 | 0.002 |

## A  Training Settings

For DeepWalk (Perozzi et al., 2014), we train Node2Vec[6] algorithm for 5 epochs with the following parameters: p=1, q=1, `walk_length`=20, `num_walks`=10, `window_size`=5.

For InfWalk[7] (Chanpuriya & Musco, 2020), a matrix factorization-based method linked to DeepWalk and spectral graph embeddings, we set the same value `window_size`=5 used for DeepWalk.

In GraphAE (Salha et al., 2020), we optimize a 1-layer GCN encoder with a random-walk loss setting analogous to DeepWalk. The model is trained for 50 iterations using Adam optimizer and learning rate of 0.01.

In GraphSAGE[8] (Hamilton et al., 2017), we optimize a 2-layer SAGE encoder with `mean` aggregation and with a random-walk loss setting analogous to DeepWalk. The model is trained for 50 iterations using Adam optimizer, learning rate of 0.01.

In DGLFRM[9] (Mehta et al., 2019), we optimize a 2-layer GCN encoder with the standard hyperparameters for variational inference: $\alpha_0$=10, $\lambda_{prior}$=0.5 and $\lambda_{post}$=1. The hidden embedding size is tuned in the list $[8, 16, 32, 64, 128, 256, 512]$. The model is trained for 300 iterations using Adam optimizer, learning rate of 0.01.

Dine[10] (Piaggesi et al., 2024), autoencoder-based post-processing process trained for 2000 iterations, and learning rate of 0.1. Input embeddings are from DeepWalk and GAE methods, tuning the input embedding size in the list $[8, 16, 32, 64, 128, 256, 512]$.

DiSe-FCAE and DiSe-GAE trained for 50 iterations using Adam optimizer and learning rate of 0.01. The hidden embedding size is tuned in the list $[8, 16, 32, 64, 128, 256, 512]$. Random walk sampling follows the same setting as DeepWalk, GraphAE and GraphSAGE.

GNNExplainer[11] is trained for 30 epochs for each test node, while PGExplainer[12] is trained for 5 epochs on trained nodes before being applied on test nodes. Moreover, since PGExplainer is based on edge masks, we derive node masks for that model with the average mask value from incident edges.

Graph explainers are applied on top of the following GNN models trained on node classification (two-layer for clique-based data and three-layer for grid-based data): GCN (Wu et al., 2019), GraphSAGE (Hamilton et al., 2017), and GATv2 (Brody et al., 2022). All the graph models (not the explainers) are tuned by searching the best output embedding size from the list $[2, 4, 8, 16, 32, 64, 128]$, as the input to the classification layer.

---

[6]https://github.com/eliorc/node2vec

[7]https://github.com/schariya/infwalk

[8]https://github.com/pyg-team/pytorch_geometric/blob/master/examples/graph_sage_unsup.py

[9]https://github.com/nikhil-dce/SBM-meet-GNN

[10]https://www.github.com/simonepiaggesi/dine

[11]https://pytorch-geometric.readthedocs.io/en/latest/generated/torch_geometric.explain.algorithm.GNNExplainer.html#torch_geometric.explain.algorithm.GNNExplainer

[12]https://pytorch-geometric.readthedocs.io/en/latest/generated/torch_geometric.explain.algorithm.PGExplainer.html#torch_geometric.explain.algorithm.PGExplainer

## B  Algorithm Complexity

Space and time complexity of DISENE can be analyzed by looking at the pseudo-code in Algorithm A1. Part of the complexity depends on the complexity of the encoder. Here, we assume GCN as encoding functions, with its own set of learnable parameters $\Theta$. But, in the experiments, we have also tested fully-connected encoders.

**Algorithm A1:** DISENE($\mathcal{G}, \mathbf{A}, K, T, L, \lambda_{\text{ent}}$)

**Input**  :  Graph $\mathcal{G} = (\mathcal{V}, \mathcal{E})$
Adjaceny matrix $\mathbf{A} \in \{0,1\}^{|\mathcal{V}| \times |\mathcal{V}|}$
Embedding size $K$, Context window $T$,
Walks length $L$, Regularization $\lambda_{\text{ent}}$
**Output:** Embedding matrix $\mathbf{H} \in \mathbb{R}^{|\mathcal{V}| \times K}$

1  Init. encoder network $Enc_\Theta(*)$;
2  Init. identity matrix features $\mathbf{X}$;
3  **while** *not converged* **do**
4       Encoding step: $\mathbf{H} \leftarrow \rho(\mathbf{W}^\top Enc_\Theta(\mathbf{A}, \mathbf{X}))$;
5       Sample batch of nodes: $\mathcal{B} \leftarrow Sample(\mathcal{V})$;
6       Init. random walks $\mathcal{W} \leftarrow \emptyset$;
7       **foreach** $v \in \mathcal{B}$ **do**
8           Sample random walk:
          $\mathcal{W} \leftarrow \mathcal{W} \cup RandomWalk(\mathbf{A}, v, L)$;
9       Random-walk loss: $\mathcal{L}_{\text{rw}}(\mathbf{H}, \mathcal{W}, T)$;
10      **foreach** $d \in \{1 \dots K\}$ **do**
11          Aggregate rows of $\mathbf{H}$: $\mathbf{f}_d \leftarrow \sum_v \mathbf{H}_{vd}$;
12          Compute 1-norm: $|\mathbf{f}|_1 \leftarrow \sum_{v,d} \mathbf{H}_{vd}$;
13      Node affiliation matrix: $\mathbf{F} \leftarrow \mathbf{H} \odot \mathbf{f}$;
14      Disentanglement loss $\mathcal{L}_{\text{dis}}(\mathbf{F})$
15      Regularization loss $\mathcal{L}_{\text{ent}}(\mathbf{F})$
16      Total loss: $\mathcal{L} \leftarrow \mathcal{L}_{\text{rw}} + \mathcal{L}_{\text{dis}} + \lambda_{\text{ent}} \mathcal{L}_{\text{ent}}$;
17      **Backpropagate and update** $\Theta, \mathbf{W}$;
18 **return** $\mathbf{H}$;

Table A2: Time and space complexity for various embedding methods in terms of number of nodes $|\mathcal{V}|$, number of edges $|\mathcal{E}|$ and latent dimensions $K$. For models trained with random walk sequences, $T$ denotes the context window size and $L$ the random walks length. For models trained with neighborhood sampling, $r$ denotes the number of sampled neighbors per node. For simplicity, in GNNs we consider single-layer architectures and we omit the $\mathcal{O}(K^2)$ memory usage of model weights since they are negligible compared to storing the embeddings.

| Method | Time Complexity | Space Complexity |
|---|---|---|
| DEEPWALK | $\mathcal{O}(|\mathcal{V}|KTL)$ | $\mathcal{O}(|\mathcal{V}|K + |\mathcal{V}|L)$ |
| INFWALK | $\mathcal{O}(|\mathcal{V}|^2 K)$ | $\mathcal{O}(|\mathcal{V}|^2)$ |
| GRAPHAE | $\mathcal{O}(|\mathcal{E}|K + |\mathcal{V}|K^2)$ | $\mathcal{O}(|\mathcal{V}|K)$ |
| GRAPHSAGE | $\mathcal{O}(r|\mathcal{V}|K^2)$ | $\mathcal{O}(r|\mathcal{V}|K)$ |
| DINE | $\mathcal{O}(|\mathcal{V}|K^2)$ +base model | $\mathcal{O}(K^2)$ +base model |
| DISENE | $\mathcal{O}(|\mathcal{E}|K + |\mathcal{V}|K^2 + |\mathcal{V}|KTL)$ | $\mathcal{O}(|\mathcal{V}|K + |\mathcal{V}|L)$ |

Our method consists of four main steps:

1. Encoding step generates the node embeddings $\mathbf{H}$ and has the same per-layer time/space complexity of standard GCNs (Duan et al., 2022), i.e. $\mathcal{O}(||\mathbf{A}||_0 K + |\mathcal{V}|K^2)$ and $\mathcal{O}(|\mathcal{V}|K)$ respectively.

2. Random walk sampling and loss calculation has time/space complexity $\mathcal{O}(|\mathcal{V}|KTL)$ and $\mathcal{O}(|\mathcal{V}|L)$ respectively (Rozemberczki et al., 2019), where $T$ is the context window size and $L$ is the random-walk length (we sample 1 random walk per node, fixing as well the number of negative samples to 1 for each positive sample). *RandomWalk* function sample a first-order random walk starting from source node $v$ of length $L$.

3. Node affiliation matrix involves computing the entries $\mathbf{F}_{ud} = \sum_{v \in \mathcal{V}_d} \phi_d(u, v; \mathbf{h})$ as $\mathbf{F}_{ud} = \sum_v \mathbf{H}_{ud}\mathbf{H}_{vd} = \mathbf{H}_{ud}\mathbf{f}_d$, i.e. by multiplying node embedding entries $\mathbf{H}_{ud}$ with quantities $\mathbf{f}_d = \sum_v \mathbf{H}_{vd}$. This step involves $\mathcal{O}(|\mathcal{V}|K)$ operations for computing and storing matrix $\mathbf{F}$.

4. Disentanglement and regularization losses involve respectively $\mathcal{O}(|\mathcal{V}|K^2)$ and $\mathcal{O}(K)$ operations for cosine similarity (matrix products) and entropy (vector sum).

Overall, given that $||\mathbf{A}||_0$ is $2|\mathcal{E}|$, DISENE results in $\mathcal{O}(|\mathcal{E}|K + |\mathcal{V}|K^2 + |\mathcal{V}|KTL)$ time complexity and $\mathcal{O}(|\mathcal{V}|K + |\mathcal{V}|L)$ space complexity. In Table A2 we compare these findings with computational complexities of competitor methods as they are reported from previous works (Tsitsulin et al., 2021; Chiang et al., 2019; Piaggesi et al., 2024), showing that our approach is in line with established node embeddings (runtime scales linearly with the number of nodes and edges).

## C Downstream Tasks Results for Real Datasets

We tested link prediction for the datasets reported in the main paper. For node classification, we tested PPI and other benchmark biological datasets in multi-label setting (Zhao et al., 2023a): the PCG dataset for the protein phenotype prediction, the HUMLOC, and EUKLOC datasets for the human and eukaryote protein subcellular location prediction tasks, respectively. Characteristics of additional biological datasets are reported in Table A3. We concatenated node attributes to node embeddings to get an enriched set of predictors that, given our method extract interpretable features, can be used in combination with feature-based explainers (e.g., SHAP) for building fully transparent prediction pipelines. In Figure A1 we report AUC-PR scores for link prediction and node classification in real-world graph data. Generally, scores increase with the number of latent embedding dimensions. Tables A4 and A5 show the maximum scores for link prediction and node classification, demonstrating that our approach can consistently achieve reasonable performances within the expected range of the performance-interpretability trade-off.

## D Explanations for Real Datasets

Figures A2 and A3 display results for Topological Alignment, Sparsity, Overlap Consistency and Positional Coherence, for real-world datasets. These curves complement the results presented in Tables 1 and 2, by covering a broader range of output embedding sizes $K$ ($K$=128 in the main paper) and tuning for the hidden size $D$, as detailed in Section A for DINE (i.e., $D \in [8, 16, 32, 64, 128, 256, 512]$). Generally speaking, DISE-GAE yields the strongest interpretability scores when the embedding dimension is large, whereas performance differences are less systematic in the low-dimensional regime. Our subsequent analysis therefore focuses on the trade-off between these interpretability metrics and downstream task accuracy (e.g., link prediction).

Figure A4 plots link-prediction AUC-PR against each interpretability metric for all the considered embedding methods. Every cluster of points represents a single method with its range of hidden and output dimensions considered as hyper-parameters. To enable a principled assessment of which method offers the most favorable balance between predictive power and transparency, the left panel of Figure A5 highlights the Pareto frontier in each scatter plot. Models on this frontier are optimal in the sense that no alternative model attains both higher accuracy and greater interpretability, therefore making any other point lying below the frontier strictly dominated with respect to the two criteria, and thus sub-optimal (dominance was assessed while incorporating the statistical uncertainty of each metric, estimated with the standard error from five independent runs per model).

Once the Pareto frontier is identified, we aim to quantify which embedding technique most consistently reaches that frontier. For every method $E$ we collect all of its hyperparameter instantiations $E = \{E_1, E_2, \dots E_m\}$ (e.g., GRAPHAE$_{K=4}$). Let $P$ be the set of all configurations that lie on the frontier. We rank methods with two complementary scores:

$$cov(E) = \frac{|P \cap E|}{|P|}, \qquad eff(E) = \frac{|P \cap E|}{|E|}.$$

Coverage is the proportion of methods in the Pareto-frontier that belongs to $E$. Efficiency is the proportion of $E$'s configurations that are Pareto-optimal. Scores reported in Table A6 combine Coverage and Efficiency via their harmonic mean to have a compact measure quantifying the "average" Pareto-optimality of every embedding method. Using this composite ranking, DISE-GAE consistently finishes within the top two methods across all datasets.

Table A3: Summary statistics of graph biological data used for multi-label node classification.

|  | PPI | PCG | HumLoc | EukLoc |
|---|---|---|---|---|
| # nodes | 3,480 | 3,177 | 2,552 | 2,969 |
| # edges | 53,377 | 37,314 | 15,971 | 11,130 |
| # labels | 121 | 15 | 14 | 22 |
| density | 0.009 | 0.007 | 0.005 | 0.003 |
| clust. coeff. | 0.173 | 0.346 | 0.132 | 0.150 |

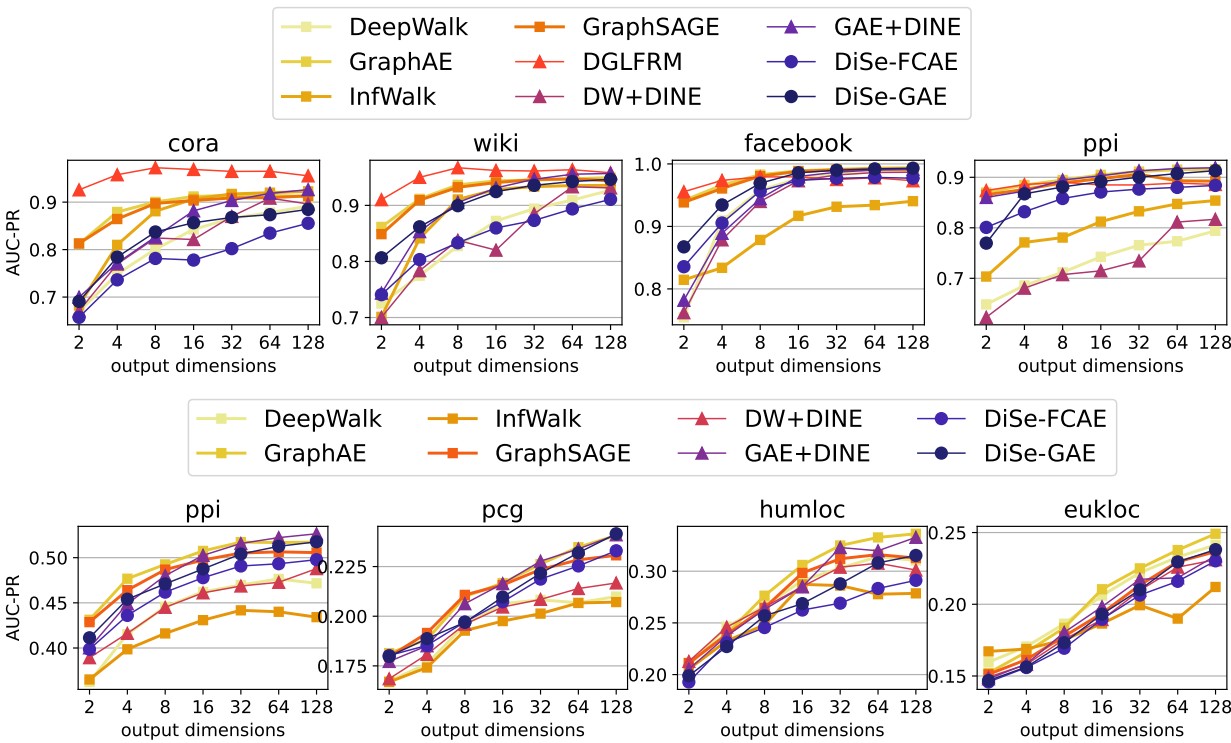

Figure A1: Downstream tasks results on real-world datasets (link prediction on the top panel, multi-label node classification on the bottom panel) with varying feature dimensions size.

Table A4: Link prediction results (AUC-PR) on real-world datasets. Best scores are in bold, while scores with a relative performance loss of no more than 2% respect to the best score are underlined.

|  | Cora | Wiki | FB | PPI |
|---|---|---|---|---|
| DeepWalk | .892±.005 | .927±.002 | .990±.001 | .794±.002 |
| GraphAE | .911±.003 | .950±.001 | **.994**±.001 | .916±.001 |
| InfWalk | .923±.003 | .936±.002 | .941±.006 | .854±.003 |
| GraphSAGE | .913±.005 | .944±.002 | .991±.001 | .892±.003 |
| DGLFRM | **.972**±.001 | **.967**±.001 | .979±.001 | .890±.001 |
| DW+Dine | .896±.004 | .931±.003 | .987±.001 | .817±.004 |
| GAE+Dine | .926±.001 | .957±.003 | .992±.002 | **.919**±.002 |
| DiSe-FCAE | .856±.007 | .911±.004 | .977±.001 | .884±.002 |
| DiSe-GAE | .885±.002 | .947±.002 | .993±.006 | .913±.001 |

Table A5: Node classification results (AUC-PR) on real-world datasets. Best scores are in bold, while scores with a relative performance loss of no more than 5% respect to the best score are underlined.

|  | PPI | PCG | HumLoc | EukLoc |
|---|---|---|---|---|
| DeepWalk | .476±.003 | .210±.001 | .314±.012 | .241±.010 |
| GraphAE | .517±.003 | .241±.001 | **.336**±.004 | **.249**±.005 |
| InfWalk | .442±.001 | .207±.002 | .287±.004 | .212±.003 |
| GraphSAGE | .506±.001 | .231±.002 | .316±.004 | .237±.011 |
| DW+Dine | .488±.002 | .217±.001 | .308±.004 | .231±.008 |
| GAE+Dine | **.526**±.001 | .241±.001 | .333±.006 | .234±.008 |
| DiSe-FCAE | .498±.001 | .233±.003 | .291±.006 | .230±.006 |
| DiSe-GAE | .518±.001 | **.242**±.004 | .315±.003 | .238±.006 |

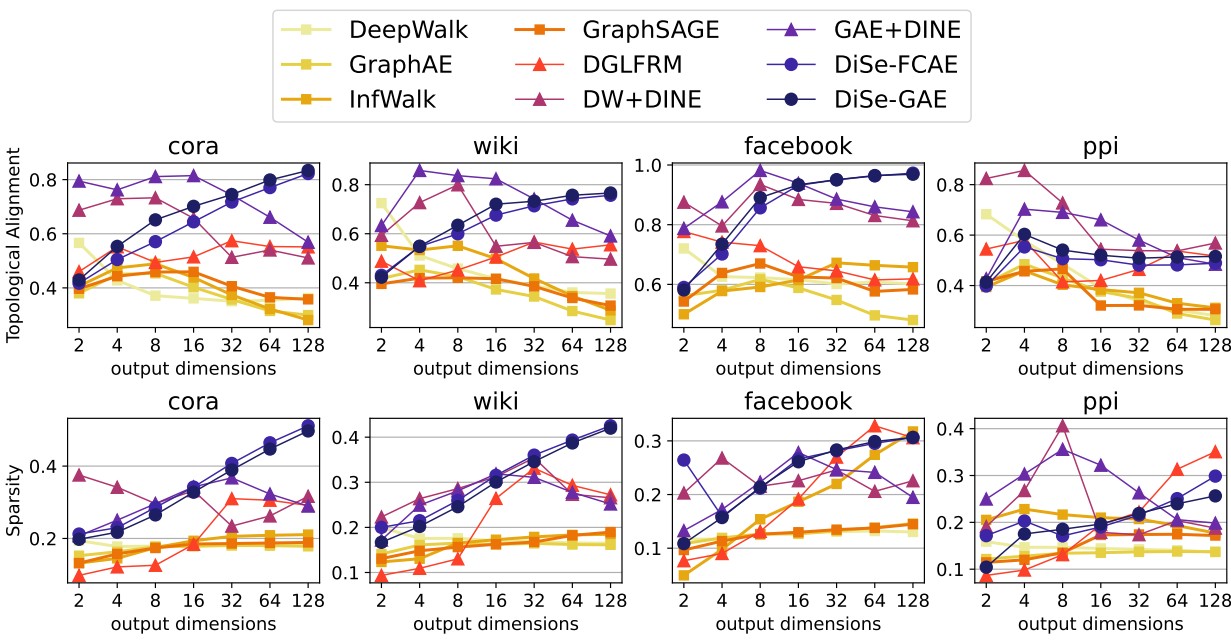

Figure A2: Topological alignment and sparsity results on real datasets with varying feature dimensions size.

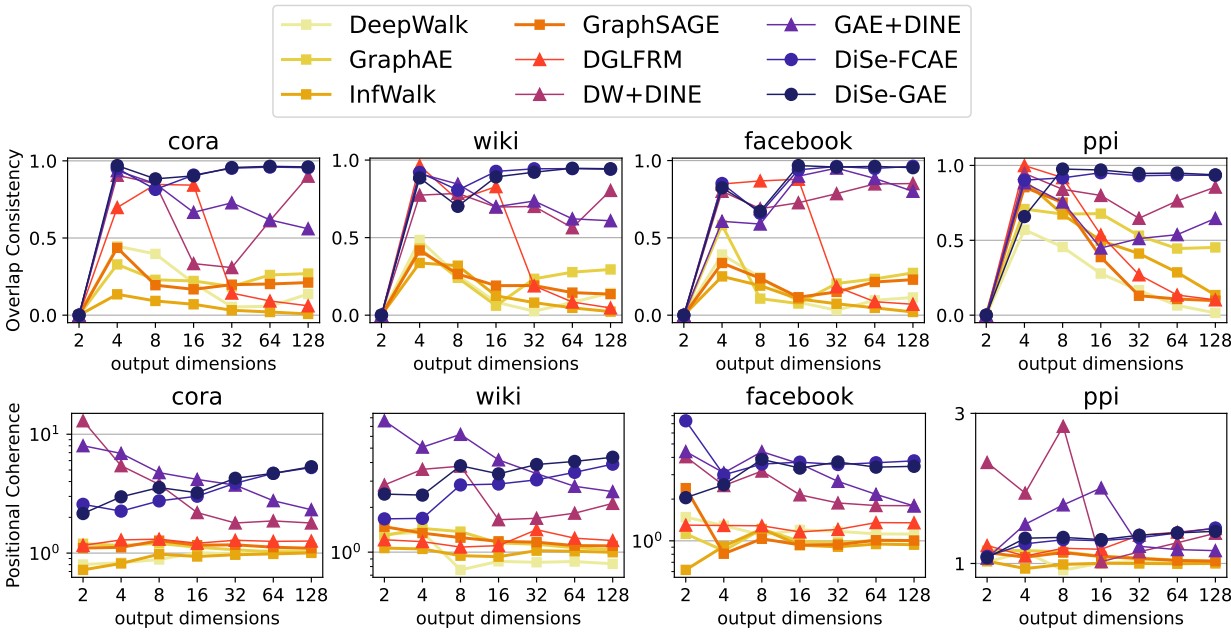

Figure A3: Overlap consistency and positional coherence results on real datasets with varying feature dimensions size.

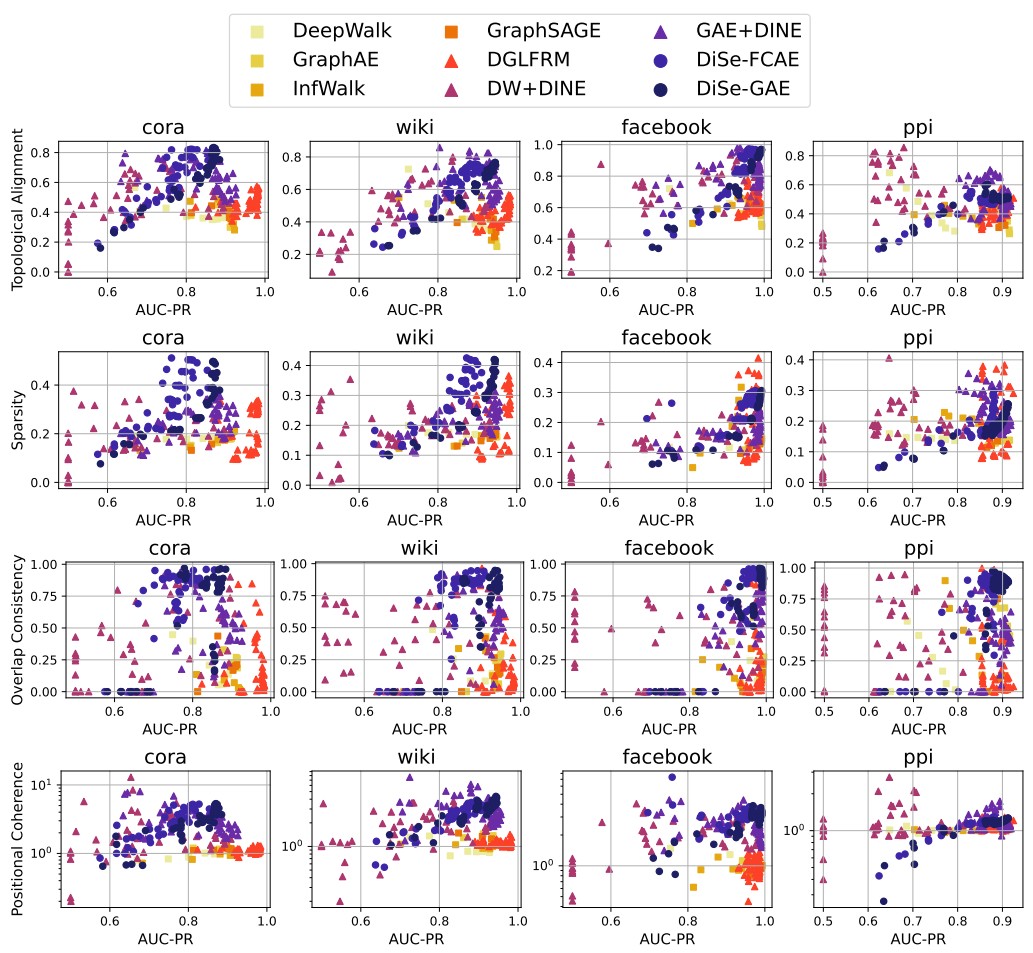

Figure A4: Scatter plots with link prediction performance (x-axis) and interpretability metrics (y-axis) on real datasets with varying feature dimensions size.

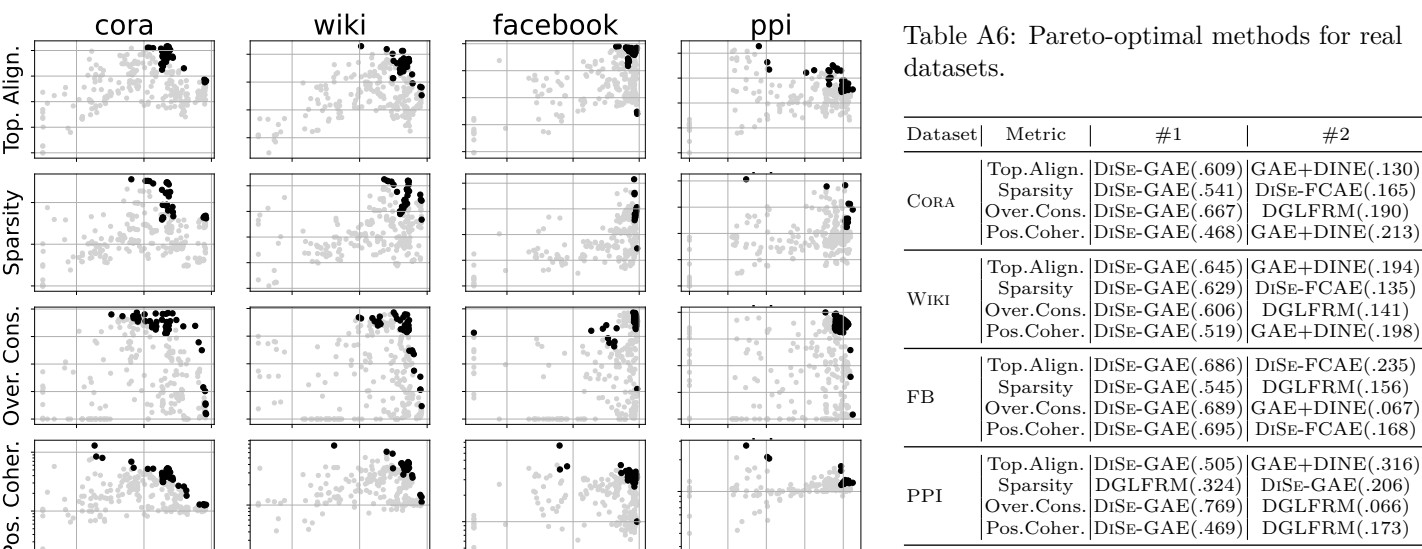

Figure A5: Pareto frontiers for the optimal models considering the interpretability-accuracy trade-offs defined by the four proposed metrics and the link prediction accuracy.

Table A6: Pareto-optimal methods for real datasets.

| Dataset | Metric | #1 | #2 |
|---|---|---|---|
| CORA | Top.Align. | DISE-GAE(.609) | GAE+DINE(.130) |
| | Sparsity | DISE-GAE(.541) | DISE-FCAE(.165) |
| | Over.Cons. | DISE-GAE(.667) | DGLFRM(.190) |
| | Pos.Coher. | DISE-GAE(.468) | GAE+DINE(.213) |
| WIKI | Top.Align. | DISE-GAE(.645) | GAE+DINE(.194) |
| | Sparsity | DISE-GAE(.629) | DISE-FCAE(.135) |
| | Over.Cons. | DISE-GAE(.606) | DGLFRM(.141) |
| | Pos.Coher. | DISE-GAE(.519) | GAE+DINE(.198) |
| FB | Top.Align. | DISE-GAE(.686) | DISE-FCAE(.235) |
| | Sparsity | DISE-GAE(.545) | DGLFRM(.156) |
| | Over.Cons. | DISE-GAE(.689) | GAE+DINE(.067) |
| | Pos.Coher. | DISE-GAE(.695) | DISE-FCAE(.168) |
| PPI | Top.Align. | DISE-GAE(.505) | GAE+DINE(.316) |
| | Sparsity | DGLFRM(.324) | DISE-GAE(.206) |
| | Over.Cons. | DISE-GAE(.769) | DGLFRM(.066) |
| | Pos.Coher. | DISE-GAE(.469) | DGLFRM(.173) |

# E   Explanations Visualization for Synthetic Datasets

**Global Subgraph Explanations**   In Figure A6 we show subgraph-level global explanations on synthetic dataset BA-Cliques. Subgraphs are generated for each feature dimension using the procedure described in Section 3.1 (summarized on the left in Algorithm A2) and are based on various unsupervised embedding methods. The explanatory subgraphs demonstrate that our method effectively aligns embedding dimensions with meaningful, non-random functional components of the graph. In contrast, standard methods such as DeepWalk and GraphAE struggle to isolate individual structural units within dimensions. Instead, their embeddings often associate dimensions with groups of cliques or subgraphs that include elements from the random Barabási-Albert scaffold. Additionally, the visualization on the right shows the correlation between latent features, further underscoring that the alignment between embedding dimensions and graph structure is closely tied to the ability to disentangle feature correlation through non-

---

**Algorithm A2:**
UnsupEdgeSubgraph($\mathcal{G}, \mathbf{Z}, d$)

**Input**  : Graph $\mathcal{G} = (\mathcal{V}, \mathcal{E})$
              Embedding function $\mathbf{z} : \mathcal{V} \to \mathbb{R}^K$
              Dimension to explain $d \in \{1 \dots K\}$
**Output** : Graph mask $\mathbf{M}^{(d)} \in \mathbb{R}^{|\mathcal{V}| \times |\mathcal{V}|}$

**1** Init. graph mask: $\mathbf{M}^{(d)} \leftarrow 0^{|\mathcal{V}| \times |\mathcal{V}|}$;
**2** Compute background average attribution:
    $\zeta_d = \frac{1}{|\mathcal{E}|} \sum_{(u,v) \in \mathcal{E}} z_d(u) z_d(v)$;
**3** **for** $(u, v) \in \mathcal{E}$ **do**
**4**     Compute edge attribution:
          $\phi_d(u, v; \mathbf{z}) = z_d(u) z_d(v) - \zeta_d$;
**5**     Add explanation:
**6**     $\mathbf{M}^{(d)}_{uv} \leftarrow \max\{0, \phi_d(u, v; \mathbf{z})\}$;
**7** **return** $\mathbf{M}^{(d)}$;

---

collinearity. Specifcially, we highlight that for DeepWalk and GraphAE the subgraphs exhibit significant overlap, which can be attributed to non-zero correlations within their latent features. In contrast, the uncorrelated features of DiSeNE produce distinct, non-overlapping explanations.

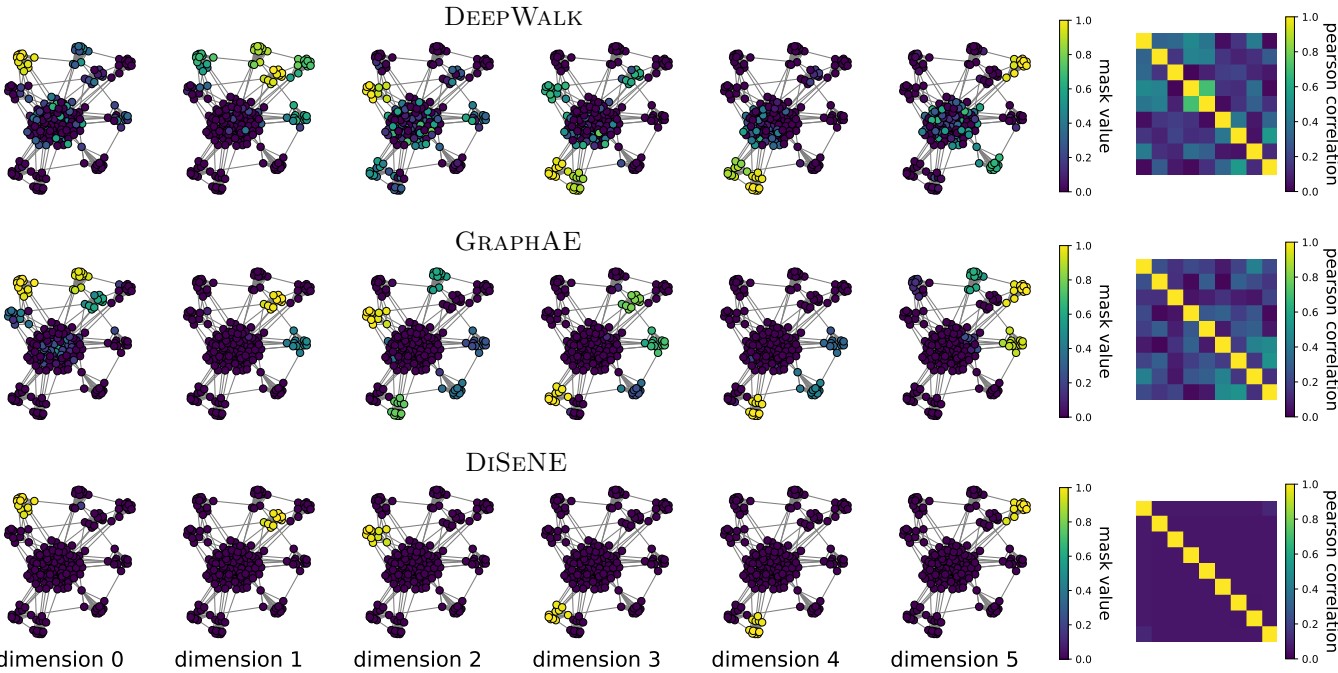

Figure A6: Subgraph-level global explanations for a representative subset of embedding dimensions, along with corresponding pairwise feature correlation plots, on synthetic dataset BA-Cliques.

**Local Subgraph Explanations**   Local explanations for node embeddings are extracted by using post-hoc feature importance method SHAP. For a given embedding model $\mathbf{h} : \mathcal{V} \leftarrow \mathbb{R}^K$ we train a downstream classifier, e.g., in node classification task or link prediction. For simplicity, here we write the case when the classifier is a (binary) linear model, but it can be any arbitrary complex model. It is anyway reasonable to assume that node embeddings come from a deep graph model and downstream classifier is a simple 1-layer neural network on top of the embedding layers.

---

**Algorithm A3:**
$\textsc{NodeClassSubgraph}(\mathcal{G}, \mathbf{\Psi}, j)$

---

**Input** : Graph $\mathcal{G} = (\mathcal{V}, \mathcal{E})$
Feature-base explanation matrix $\mathbf{\Psi} \in \mathbb{R}^{|\mathcal{V}| \times K}$
Dimension to explain $j \in \{1 \dots K\}$
**Output** : Node mask $\mathbf{B}^{(j)} \in \mathbb{R}^{|\mathcal{V}|}$

**1** Init. node mask: $\mathbf{B}^{(j)} \leftarrow 0^{|\mathcal{V}|}$;
**2** **for** $v \in \mathcal{V}$ **do**
**3**   Add explanation:
**4**   $B_v^{(j)} \leftarrow \max\{0, \Psi_j(v; b)\}$;
**5** **return** $\mathbf{B}^{(j)}$;

---

$$\text{(node classification)} \quad b(v) = \sigma(\sum_{j=1}^{K} \beta_j h_j(v) + \beta_0)$$

$$\text{(link classification)} \quad b(u, v) = \sigma(\sum_{j=1}^{K} \beta_j h_j(u, v) + \beta_0)$$

Given a vector representation of a graph instance (e.g., a node embedding $\mathbf{h}(v)$ or an edge embedding $\mathbf{h}(u, v)$), and the corresponding prediction from classifier $b$, we compute feature importance with SHAP $\{\Psi_j^{(\mathcal{V})}(v; b)\}_{j=1\dots K}$ or $\{\Psi_j^{(\mathcal{E})}(u, v; b)\}_{j=1\dots K}$ and the corresponding task-based graph masks (we illustrate the pesudo-code for node classification masks in Algorithm A3):

$$\mathbf{B}^{(j)}(\mathbf{\Psi}^{(\mathcal{V})}) \in \mathbb{R}^{|\mathcal{V}|}; \quad B_v^{(j)} = \max\{0, \Psi_j^{(\mathcal{V})}(v; b)\}$$

$$\mathbf{B}^{(j)}(\mathbf{\Psi}^{(\mathcal{E})}) \in \mathbb{R}^{|\mathcal{V}| \times |\mathcal{V}|}; \quad B_{uv}^{(j)} = \max\{0, \Psi_j^{(\mathcal{E})}(u, v; b)\}$$

It is valuable to remark that, training with logistic regression and applying SHAP, the resulting importance scores are simply the coefficients of the regression (Lundberg & Lee, 2017) $\Psi_j(x; b) = \beta_j(h_j(x) - E[h_j])$. Thus, combining this methodology to interpretable graph features of DiSeNE, we obtain a fully transparent node/edge classification pipeline for graph data.

Figure A7 present examples of local explanations for node classification tasks on the small-sized synthetic datasets Ba-Cliques and Tree-Grids, using different methods. The experimental settings are consistent with those described in the main paper. On the left, we highlight the local ground-truth structures for the instance nodes depicted in the illustrations. On the right, we display the explanation subgraphs generated by each method, with nodes color-coded according to the respective explanation masks. For GraphAE and DiSeNE, the visualized subgraphs represent the most relevant structures as determined by feature importance attribution from the logistic regression classifier. For GNNExplainer and PGExplainer, the node masks correspond to the algorithm's output in explaining a 2-layer GCN (Wu et al., 2019). Notably, DiSeNE demonstrates a strong ability to produce meaningful and interpretable node masks, effectively competing with state-of-the-art GNN explanation methods.

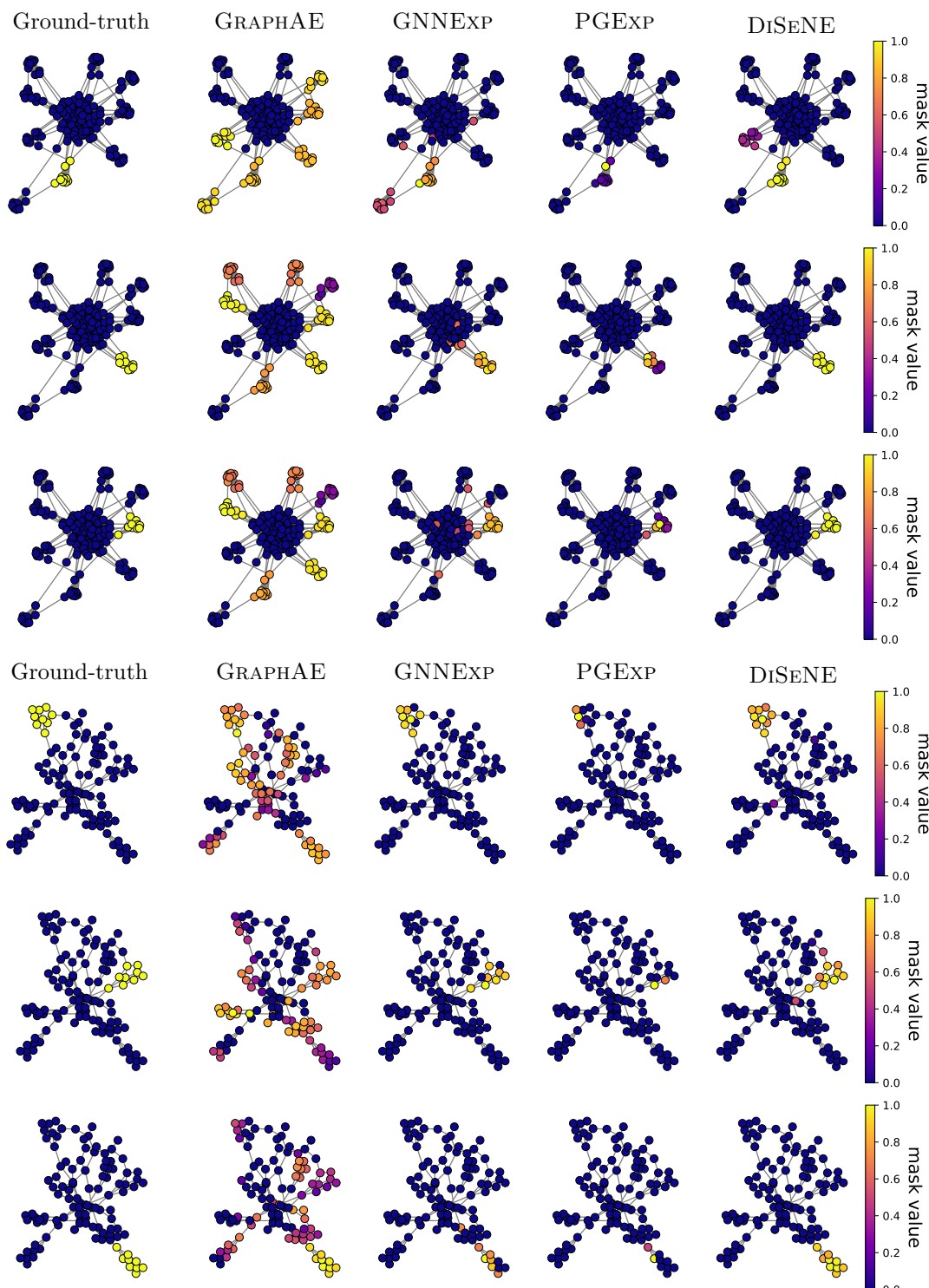

Figure A7: Subgraph local explanations for node classification in BA-CLIQUES (top) and TREE-GRIDS (bottom). On the leftmost column, we highlight the local ground-truth structures for the considered instance nodes. On the other columns, we display the explanation subgraphs generated by each method, with nodes color-coded according to the respective explanation masks. For GRAPHAE and DISENE, the visualized subgraphs represent the most relevant structures extracted with Algorithm A3 and determined by feature importance attribution from the logistic regression classifier. For GNNEXPLAINER and PGEXPLAINER, the node masks correspond to the algorithm's output in explaining a GCN (Wu et al., 2019) in node classification.

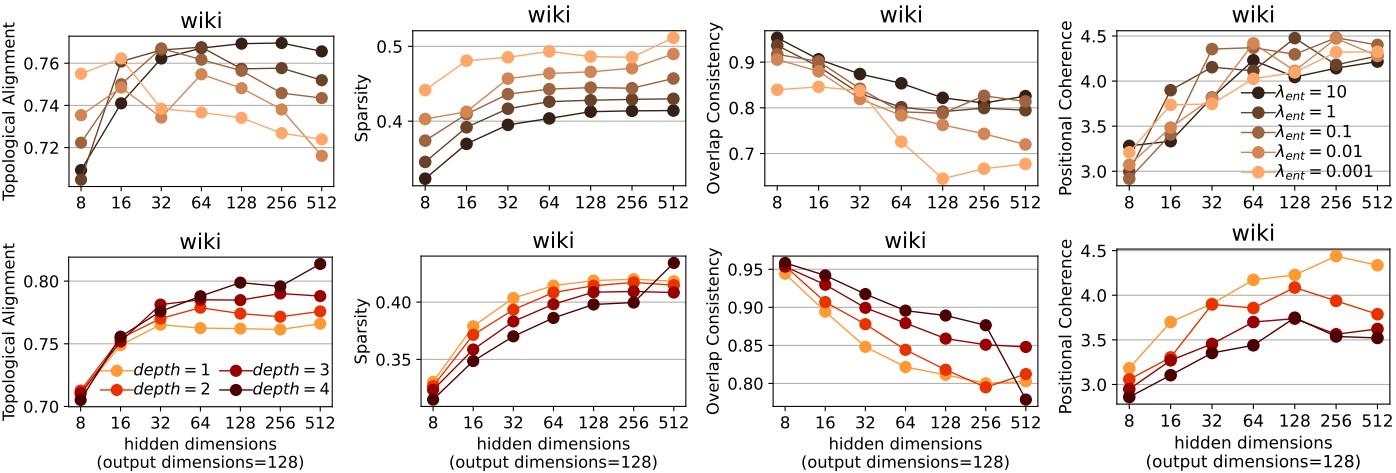

Figure A8: Interpretability metrics computed in WIKI with varying entropy regularization parameters, number of convolutional layers and hidden dimension sizes in DISE-GAE.

## F    Explanations for Synthetic Datasets

**Global Explanations**   In Figure A9 we plot results for Topological Alignment and Sparsity, on the top and the bottom respectively, on synthetic datasets. Generally, DISE-FCAE outperforms DISE-GAE and the other competitors in all the datasets. In Figure A10 we plot results for Overlap Consistency and Positional Coherence, on the top and the bottom respectively, on synthetic datasets. For the overlap metric, DISE-FCAE and DISE-GAE consistently outperform the competitors, especially with more than 8 dimensions where they achieve almost perfect overlap. For the positional metric, the competitors GAE+DINE and DW+DINE slightly outperform DISE methods, especially in large dimensions, while DEEPWALK also show good results.

**Local Explanations**   In Figure A11 we plot results for the plausibility metric on link prediction and node classification, on the top and the bottom respectively, while comparing different unsupervised methods that output node embeddings. Plausibility seems to benefit larger dimension values for DISE methods and DW+DINE for link prediction. Figure A12 shows the corresponding downstream task accuracy results.

## G    Influence of Entropy Regularizer and Model Depth for Wiki Dataset

Figure A8 explores how both the entropy-regularization coefficient, $\lambda_{ent}$, and the GCN depth in DISE-GAE influence the interpretability metrics on the WIKI dataset. Increasing $\lambda_{ent}$ and using deeper networks boost Topological Alignment and Overlap Consistency, but it also reduce Sparsity, revealing a trade-off between compactness and explanatory power. Positional Coherence is weakly sensitive to $\lambda_{ent}$, and tends to improve with shallower architectures.

## H    Qualitative Examples for Cora Dataset

Tables A7 and A8 presents qualitative examples regarding the embedding interpretability on the CORA citation graph. The dataset comprises seven classes, each corresponding to a distinct computer science topic. We employ different embeddings as predictor variables for the node classification task with a one-vs-rest multiclass strategy using Logistic Regression. Comparing DISE-GAE with the vanilla GAE and GAE+DINE, we selected for each class the most task-relevant embedding dimension corresponding to the highest coefficient of the linear classifier. For every selected dimension, we analyze different properties of the corresponding anchor subgraph, computed with Eq. (2). First, without inspecting node class labels, the Topological Alignment scores and the Density scores (i.e., the ratio of the number of edges with respect to the maximum possible edges in the subgraph) tell us that DISE-GAE produces denser subgraphs compared to the overall

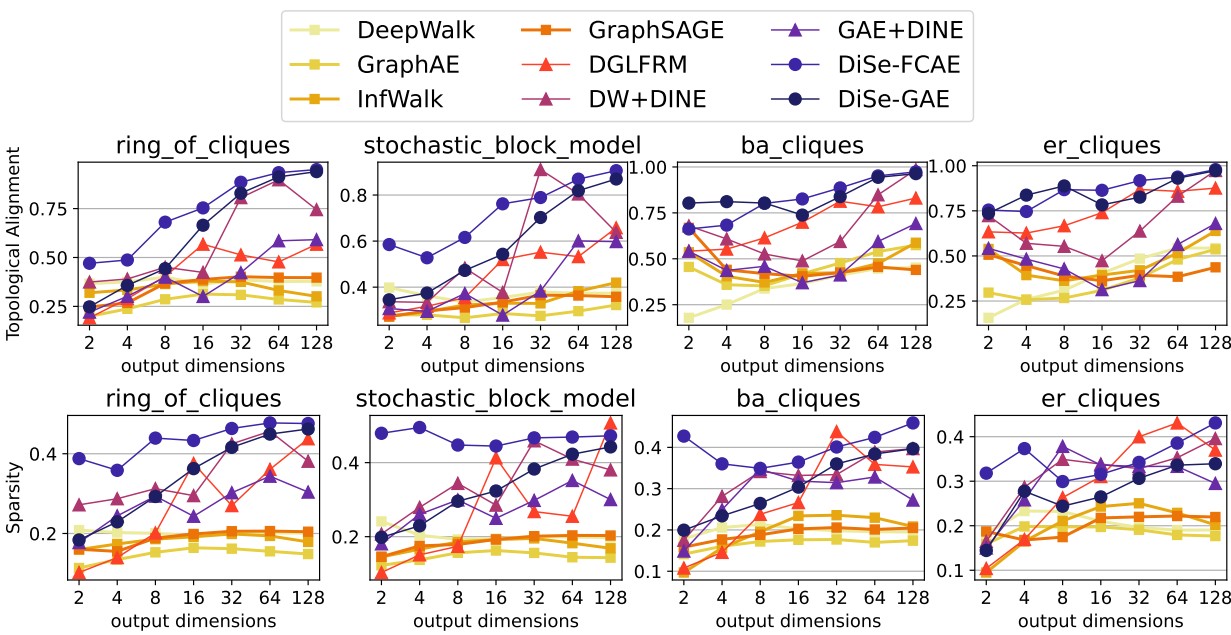

Figure A9: Topological alignment and sparsity results on synthetic datasets with varying feature dimensions size.

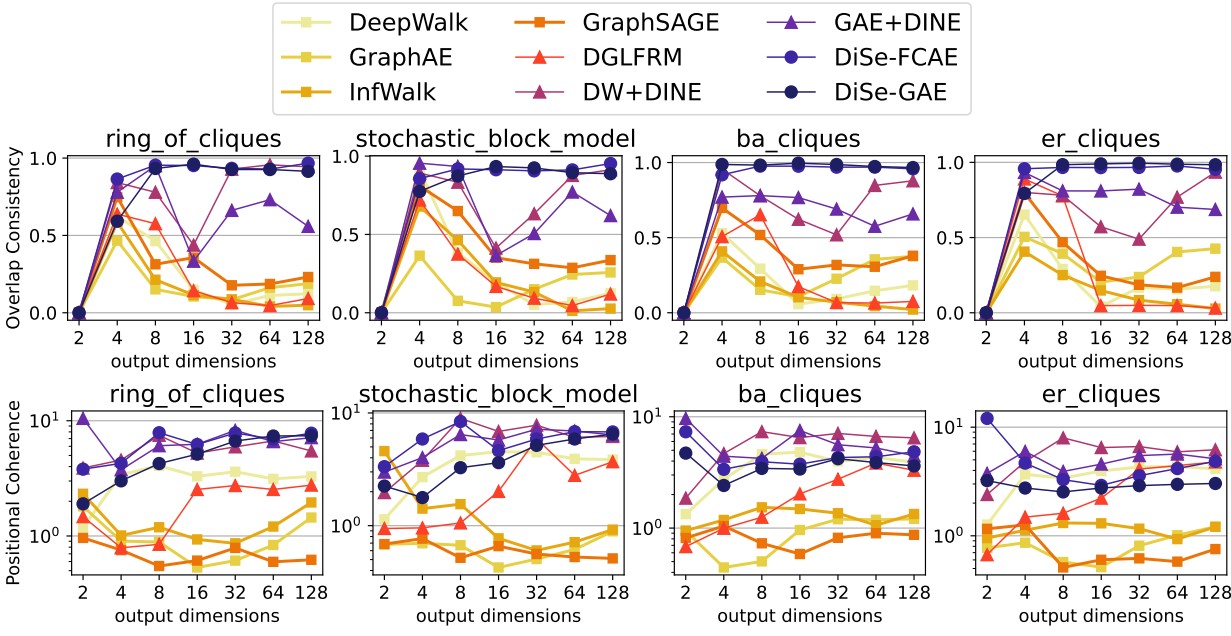

Figure A10: Overlap consistency and positional coherence results on synthetic datasets with varying feature dimensions size.

graph, indicating stronger structural cohesion in the most task-relevant subgraph. Moreover, considering the bar plots that display distributions of class labels inside the subgraphs, nodes grouped within a single dimension of DISE-GAE share research themes more consistently. This is quantified in the lower entropy of labels' distributions within each subgraph. Furthermore, by looking at the top-20 most frequent keywords extracted from each cluster (after lemmatization and stop-words removal), we represent the word-cloud for each dimension, indicating clusters that are more thematically coherent and easier to interpret. For example,

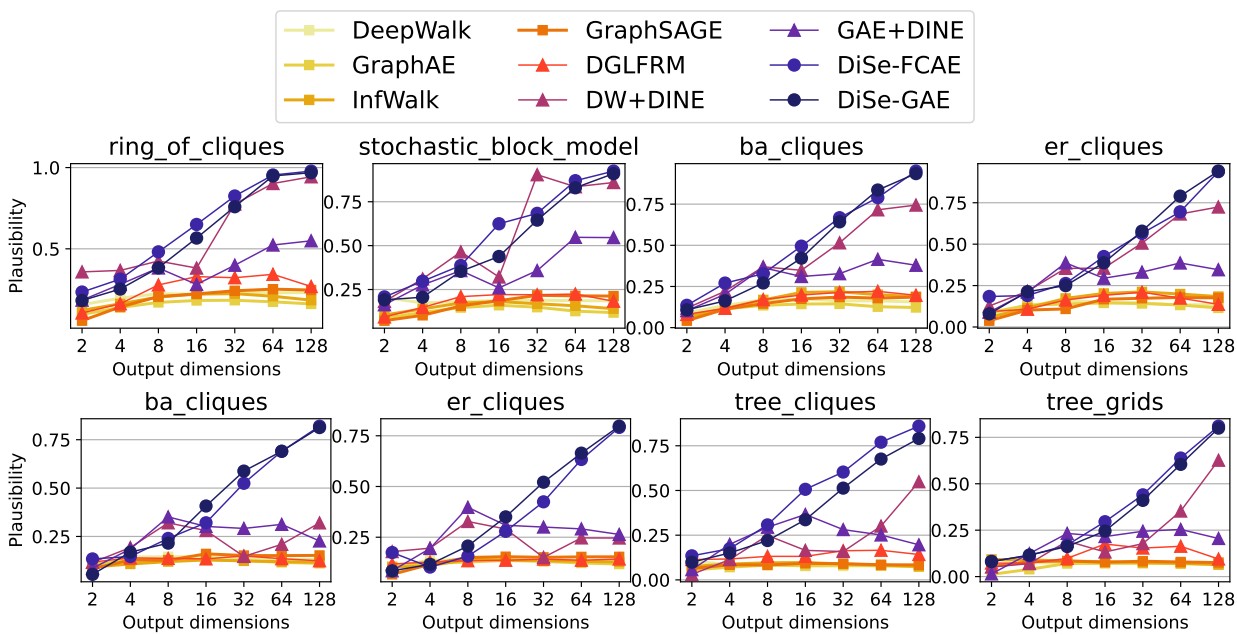

Figure A11: Plausibility results on synthetic datasets (link prediction on the top panel, binary node classification on the bottom panel) with varying feature dimensions size.

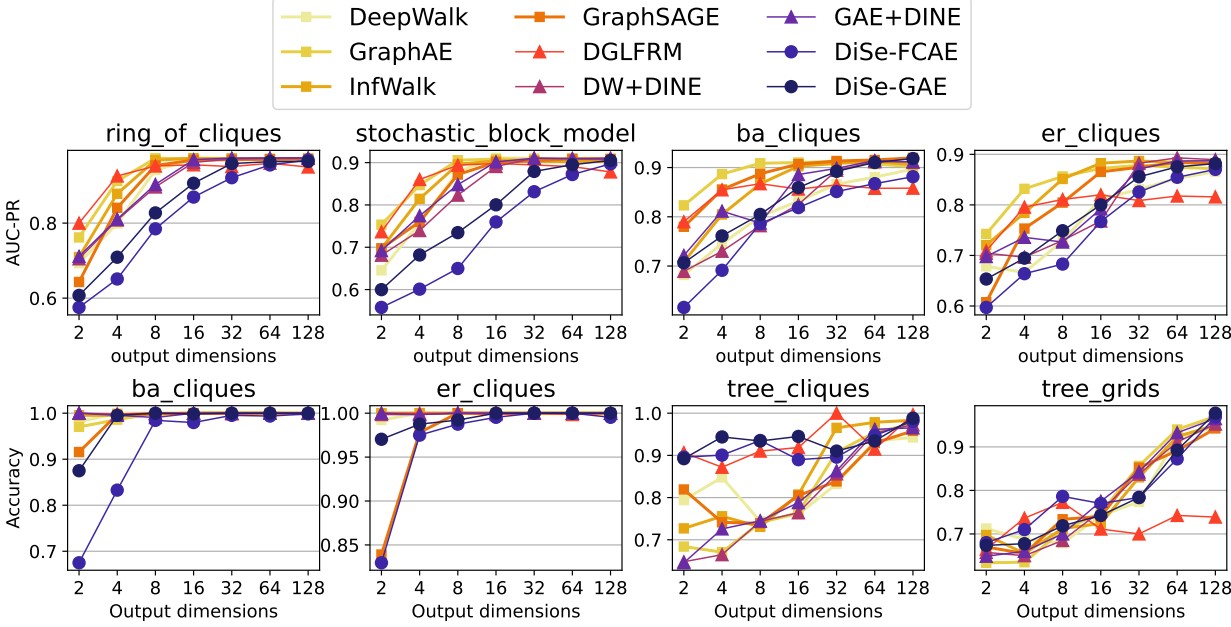

Figure A12: Downstream tasks results on synthetic datasets (link prediction on the top panel, binary node classification on the bottom panel) with varying feature dimensions size.

looking at the most predictive dimensions for the label *Probabilistic methods*, we observe with DISE-GAE we identify papers mentioning topic-relevant keywords such as "bayesian netwoks", "inference" and "probabilistic"; while papers identified with GAE+DINE frequently refer to "reinforcement learning" and "control". These qualitative examples therefore show that our model learns latent dimensions whose structural and semantic meanings are markedly more human-intelligible than those of the competing methods.

Table A7: Most task-relevant dimensions in different methods for node labels *Neural Networks*, *Reinforcement Learning*, and *Probabilistic Methods* and their interpretations based on topic and text information in CORA.

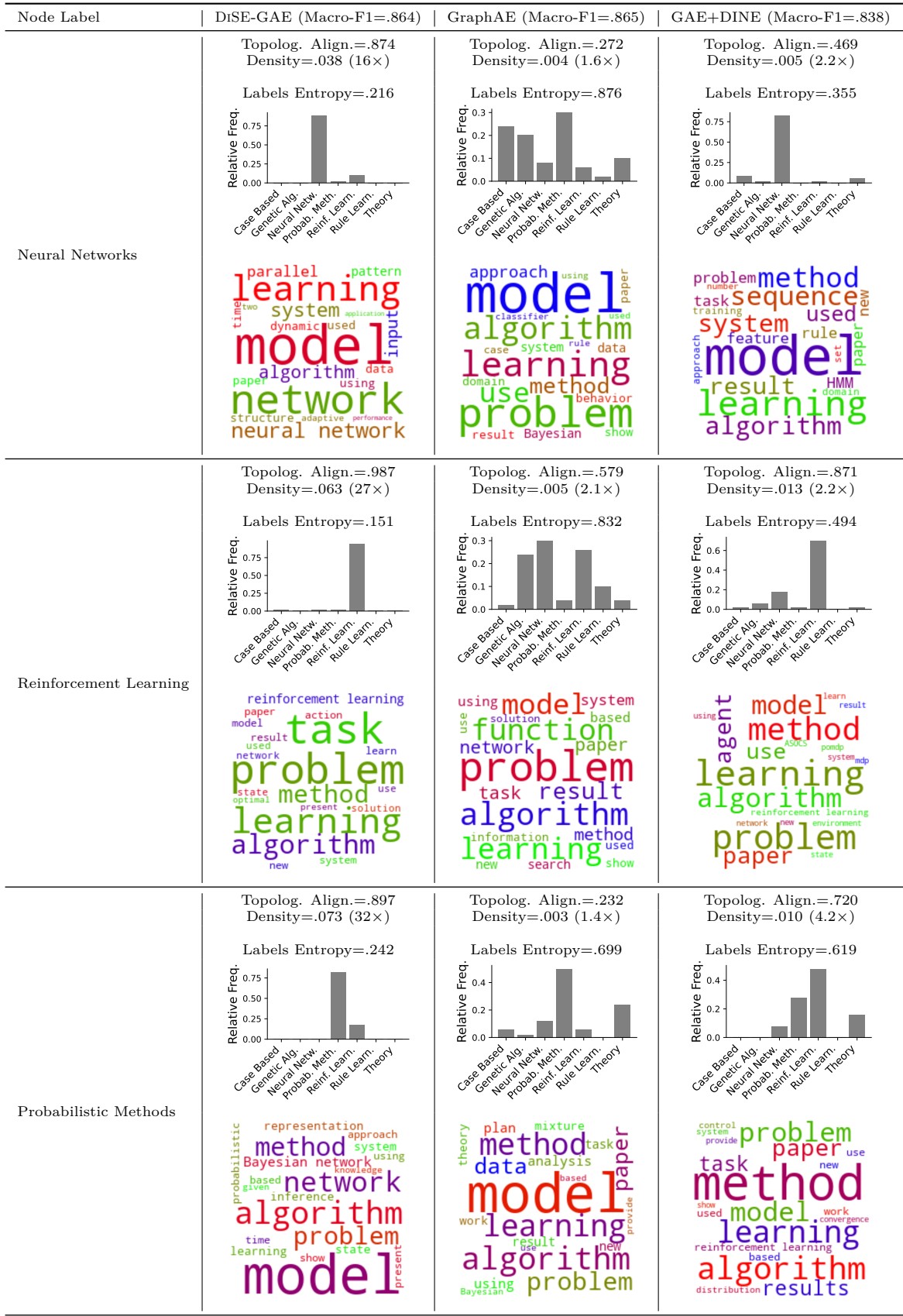

Table A8: Most task-relevant dimensions in different methods for node labels *Theory*, *Genetic Algorithms*, and *Rule Learning* and their interpretations based on topic and text information in CORA.

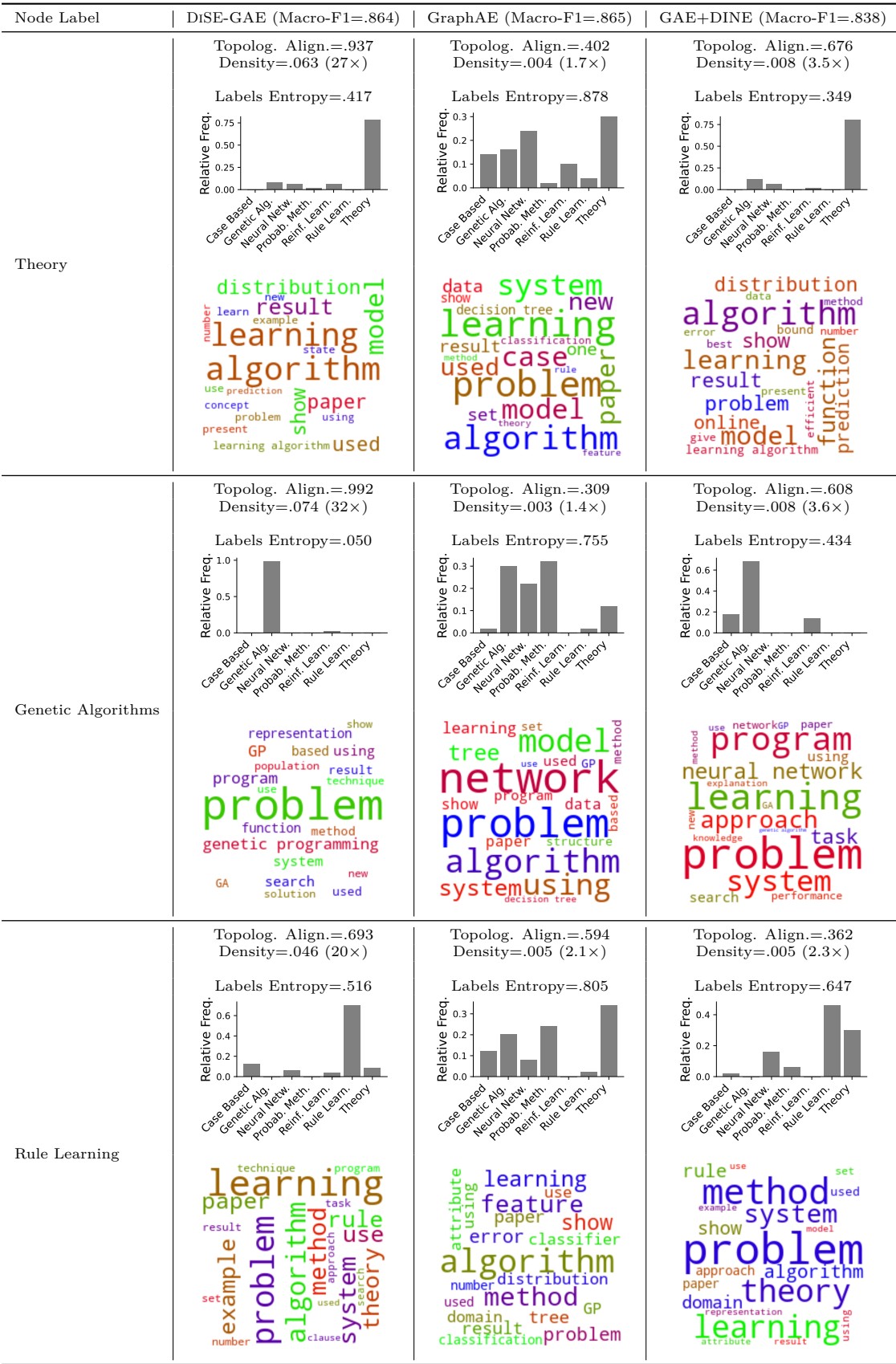

