# OpenReview forum: "Disentangled and Self-Explainable Node Representation Learning"
_TMLR — Accepted by TMLR_

### Review · Reviewer_nkUJ · 2025-04-29

**Summary Of Contributions:**

This paper presents DISENE, a novel framework for learning self - explainable node embeddings in an unsupervised way. It decouples representation learning and interpretability objectives, enabling each embedding dimension to correspond to a unique topological sub - structure. With three key objectives (connectivity preservation, dimension interpretability, and structural decoupling) and a comprehensive loss function, it ensures high - quality embeddings. Experiments on real - world and synthetic datasets show that DISENE excels in human understandability, structural decoupling, and downstream task utility, while also introducing new metrics for evaluation.

**Audience:**

Yes

**Claims And Evidence:**

Yes

**Requested Changes:**

Please refer to the weaknesses section.

**Strengths And Weaknesses:**

Strengths：
1. By decoupling representation learning and interpretability objectives, DISENE enables each embedding dimension to correspond to a unique topological sub-structure, which addresses the lack of interpretability in existing unsupervised node embedding methods.
2. DISENE shows excellent performance on multiple real-world and synthetic datasets in terms of human understandability, structural decoupling, and downstream task utility. The high scores in understandability, overlap consistency, and credibility in the experiments demonstrate its effectiveness.
3. The introduction of new metrics helps to evaluate the understandability of explanations and analyze the impact of spatial structure and node position on latent features, providing a more comprehensive understanding of the method.

Weaknesses:
1. The results are based on specific experimental settings, such as the fixed output embedding dimension in real-world data and the range of dimensions in synthetic data. It is unclear how the method performs under different or more complex experimental settings, and there may be limitations in generalization.
2. Although new metrics are introduced, the understanding and evaluation of interpretability are still subjective to some extent. There may be other aspects of interpretability that are not fully captured by the current metrics, and further research is needed to improve and expand the interpretability evaluation system.
3. The framework involves multiple objectives and constraints, such as the random walk optimization framework, calculation of edge importance, and soft orthogonality constraints. This may lead to relatively high computational complexity, especially for large-scale graphs, which could limit its scalability in real-world applications.

---

> ### Author Response · Authors · 2025-05-14
> **Response to Reviewer nkUJ**
>
> We appreciate the reviewer’s constructive feedbacks. Here we provide detailed answers to the specific concerns.
>
> **W1.** In the original manuscript, we mainly focused on understanding the role of the embedding dimension as the main hyperparameter because prior studies identify it as the most influential factor for embedding models [1-3]. Moreover, we limited our experiments on 128-dim embeddings for real data because, in practice, high dimensionalities typically saturates link prediction performance, thus yielding the most accurate graph representations. Reporting results under these conditions therefore allowed us to showcase the methods at their optimal accuracy, where interpretability has more value (e.g., it would be useless to have a fully interpretable, yet inaccurate model). For synthetic data instead, we made an exhaustive evaluation given the controlled scenario and the small-scale setting.
> However, we recognize the value of analyzing more in-depth the full spectrum of trade-offs between accuracy and interpretability. To address the comment, we first report in Appendix D Figures A2–A3 of the revised manuscript the full set of results obtained with multiple embedding dimensions on the real datasets. Moreover, in Figure A4 of the revised version, we plot link prediction AUC-PR against each interpretability metric for every embedding, in order to include in the analysis every method instantiation across the complete hyperparameter grid. Each cluster of points represents one method, evaluated over the explored ranges of hyperparameters. In Figure A5 (revised manuscript), we highlight
> the Pareto frontier for each scatter plot and rank every method according to its presence on that frontier (the Pareto sets may appear denser than expected because we account for statistical errors resulting from multiple embedding replications for each model). In Table A6 (revised manuscript), we show that \textsc{DiSe}-GAE offers the best trade-off between accuracy and interpretability. Taken together, these expanded experiments and the Pareto-based analysis reveal consistent patterns that, in our view, strengthen the generality of the results.
>
> [1] Nakis, Nikolaos, et al. "How Low Can You Go? Searching for the Intrinsic Dimensionality of Complex Networks using Metric Node Embeddings." ICLR, 2025.
>
> [2] Chanpuriya, Sudhanshu, et al. "Node embeddings and exact low-rank representations of complex networks." NeurIPS, 2020.
>
> [3] Yin, Zi, and Yuanyuan Shen. "On the dimensionality of word embedding." NeurIPS, 2018.
>
> **W2.** We agree with the reviewer’s comment that there is always room for future work to improve and expand the graph embedding interpretability evaluation system. Anyway, we view this not as a limitation but as a strength of our work: having shed light on novel aspects of interpretability that have been largely overlooked in previous studies, our work opens new avenues for advancing human-centric and human-in-the-loop applications in graph machine learning research.
>
> **W3.** We appreciate the reviewer's concern highlighting the importance of computational efficiency. We have already proved in Appendix B of the original manuscript that the overall time and space complexities scale linearly with the number of nodes and edges in the graph.
> Moreover, when working with large-scale graphs, the method fully supports node mini-batch strategies and other approaches for improving efficiency (e.g., subgraph sampling [4]) which further enhance scalability. In the revised manuscript, we have added Appendix Table A2 that compares time and space complexity of our model against baseline methods, showing our approach remains competitive on both runtime and memory usage.
>
> [4] Salha, Guillaume, et al. "Fastgae: Scalable graph autoencoders with stochastic subgraph decoding." Neural Networks, 2021.

---

### Review · Reviewer_dLkq · 2025-04-30

**Summary Of Contributions:**

The paper presents a stochastic block model (SBM) motivated graph model that attempts to disentangle the node embedding, unveil the underlying community and improve interpretability. The node representation is encoded by GNN, and loss terms are proposed to better learn the block model.

**Audience:**

Yes

**Claims And Evidence:**

Yes

**Requested Changes:**

Please refer to the weakness part and answer the questions. The authors are supposed to compare the mentioned interpretable probabilistic graph generation model that captures the community structure and learns interpretable node representations.

**Strengths And Weaknesses:**

Strengths:
1. The writing is clear and easy to understand, the notations and figures are presented properly.
2. The proposed evaluation metric looks interesting to me, and can be considered as useful information to this community in the future.

Weaknesses:
1. From the statement made above Eq(1), how could the authors assume that each dimension $d$ of the node representation is independent or weakly dependent such that each dimension can be evaluated separately, and contribute to the likelihood?
2. It is acceptable if the authors treat $\phi_d(\cdot)$ as a score metric, but in the training, a non-linear activation is applied on, so I am concerned about whether it can be simply used as a valid metric for evaluating dimension importance (Eq(2)) like the other linear model.
3. The dimension of node embedding, $d$, which also determines the number of global edge mask $M^{d}$ is determined by the graph structure such that it is data-dependent, how to choose it is empirical.
4. The 'node affiliation matrix' $F$ is essentially a block model to me that captures the community correlation underlying the whole graph. This is a well-studied problem and approaches of explicitly learning the block model can be found in [1,2]. The authors are expected to discuss and compare with the closely related works, which use a probabilistic model and explicitly learns the block model which was shown to capture the community structure.
5. I am confused why in entropy model $H$ the numerator is not applied absolute value. Also, the intuition of using entropy model here is not clear to me, since maximizing the entropy will force each dimension to be equally learned (assume the embeddings are non-negative at this point), but the goal of this model is to disentangle the dimension, which contradicts with each other.
6. The human understandability can be improved, especially when the dataset (CORA, WIKI) used has rich topic and text information, the authors are expected to show more on how the learned node representation is interpretable.

[1] Mehta, Nikhil, Lawrence Carin Duke, and Piyush Rai. "Stochastic blockmodels meet graph neural networks." International Conference on Machine Learning. PMLR, 2019.

[2] Chen, Xiaohui, Xi Chen, and Liping Liu. "Interpretable Node Representation with Attribute Decoding." Transactions on Machine Learning Research, 2022.

---

> ### Author Response · Authors · 2025-05-14
> **Response to Reviewer dLkq (Part 1)**
>
> We appreciate the reviewer’s constructive feedbacks. Here we provide detailed answers to the specific concerns.
>
> **W1-2.** We agree that the statement made above Eq(1) needs further clarification, since we realise that our wording gave the impression that we require independence among embedding dimensions. The likelihood in Eq(1) does not need that assumption; it only needs the additive structure of the dot-product decoder that is standard in graph-embedding models. We clarify this below (we added this explanation to the revised manuscript):
>
> - **Additive contributions:** In the likelihood score $\hat{y}(u,v)$ we have inside the sigmoid a linear combination of the per-dimension products $z_d = h_d(u) h_d(v)$. Because the sigmoid is monotone, analyzing the contribution of each $z_d$ is sufficient to explain the final likelihood. No statistical dependence between $z_d$ terms is required for this decomposition, only linear additivity.
>
> - **Attribution is meaningful even with correlated dimensions:** importance scores $\phi_d$ quantify how much above or below its typical value dimension $d$ pushes the sigmoid for a specific edge. If two dimensions are correlated, they may often push an edge in the same direction, but each still provides an observable marginal deviation that we can measure in this way. In other words, $\phi_d$ quantifies marginal responsibility, not exclusive responsibility; therefore, it remains valid in the presence of correlations.
>
> **W3.** We currently treat the embedding dimension $d$ as a tunable hyperparameter. We agree that a more principled procedure should take into account graphs’ intrinsic dimensionality [1,2], to obtain even more meaningful models and interpretations. However, developing such an adaptive dimension-selection strategy lies outside the scope of this paper, but we recognise its importance and plan to pursue it in future work.
>
> [1] Nakis, Nikolaos, et al. "How Low Can You Go? Searching for the Intrinsic Dimensionality of Complex Networks using Metric Node Embeddings." ICLR, 2025.
>
> [2] Gu, Weiwei, et al. "Principled approach to the selection of the embedding dimension of networks." Nature Communications, 2021.
>
> **W4.** We agree that our affiliation matrix approach, at first glance, resembles a SBM membership matrix, because both describe how nodes participate in mesoscale structures. However, our approach differs from the SBM-based GNN methods in several important ways:
>
> - **Modeling paradigm:** SBM-based GNNs are probabilistic generative models that posit priors over community memberships and edge probabilities and then perform variational MLE. Our method employs a discriminative model that learn embeddings with random walk classification loss and makes no parametric assumptions on the edge-generation process beyond the observed sparsity.
>
> - **Optimized variables:** in SBM-based GNNs the community affiliations are optimized as explicit parameters, while in our framework the membership matrix is a derived quantity from embedding variables and it is used to impose disentanglement.
>
> - **Granularity of interpretability:** SBM-based GNNs can explain the graph at the community-level and require external semantic labels to be meaningful, similarly to our new qualitative analysis in Appendix H (e.g., by ranking or filtering nodes according to community strengths and checking whether these groups align with semantic classes). Our method is grounded on dimension-level interpretability,  so each latent coordinate aligns with an anchor subgraph; a practitioner can inspect a single dimension and immediately see which specific edges it explains.
>
> - **Use of node attributes:** the referenced methods take into account node attributes, possibly confounding topology with semantics, while our method is solely based on graph structure.
>
> We hope this makes the distinction between our method and SBM-based GNNs clear.
> We extended the benchmark experiments to include the Deep Generative Latent Feature Relational Model (DGLFRM) [3], using the authors’ public implementation (no publicly available code was found for [4]). DGLFRM now appears as an additional baseline in every table and figure. It has competitive scores on Sparsity across both synthetic and real-world data sets;  additionally, Comprehensibility (renamed Topological Alignment) and Positional Coherence fall in an intermediate band, better than fully dense embeddings but worse than dimension-based interpretable ones. However,  its performance on Plausibility and Overlap Consistency is markedly poor. These results highlight the qualitative gap between our approach and SBM–based GNNs such as DGLFRM.
>
> [3] Mehta, Nikhil, Lawrence Carin Duke, and Piyush Rai. "Stochastic blockmodels meet graph neural networks." ICML, 2019.
>
> [4] Chen, Xiaohui, Xi Chen, and Liping Liu. "Interpretable node representation with attribute decoding." TMLR, 2022.

---

> ### Author Response · Authors · 2025-05-14
> **Response to Reviewer dLkq (Part 2)**
>
> **W5.** We agree with the reviewer’s concern regarding the use of entropy regularization. Because every node feature $h_d(u)$ is taken after the encoder’s ReLU activation, the aggregated values $\sum_d h_d(u)$ are already non-negative; hence an absolute value is unnecessary. In the revised manuscript we have added a footnote clarifying this assumption. Moreover, we clarify that the use of entropy regularizer complements, rather than contradicts, the disentanglement objective. The loss in Eq(3) encourages columns of matrix $F \in \mathbb{R}^{V \times K}$ to be orthogonal. Orthogonality alone, however, admits a degenerate but valid optimum in which one (or several) columns are identically zero. Such “empty” factors carry no information and hurt interpretability. Specifically, as correctly argued by the reviewer, regularization is necessary to guarantee that each dimension is equally learned. In the top panel of Figure A8 (in the revised version) we empirically analyze the influence of the entropy regularization in our model. We observe that entropy regularization positively impacts overlap consistency, which is closely tied to disentanglement.
>
> **W6.** We appreciate the reviewer’s suggestion on illustrating concrete examples that combine graph structure with textual and topic information available in graph data. Tables A7-A8 in Appendix H now presents the most task-relevant embedding dimensions on the CORA dataset, evaluated through multi-class logistic regression, together with their associated keywords and class labels distributions. The anchor subgraphs extracted by our method are more interpretable: they are internally denser and more aligned with meaningful meso-scale structures. Moreover, nodes clustered within a single dimension share research themes more consistently than those produced by the baselines; this is also reflected in the word-clouds of each cluster. Overall, the new qualitative examples highlight that our approach obtains dimensions whose structural and semantic meanings are more human-understandable.

---

### Review · Reviewer_j9WJ · 2025-04-30

**Summary Of Contributions:**

This paper proposes DiSeNE (Disentangled & Self-Explainable Node Embedding), a framework that learns embeddings where each latent dimension aligns with a specific topological sub-structure (a global “anchor” subgraph) of the input graph. To obtain such embeddings, the authors propose to jointly optimize a random-walk reconstruction loss $\mathcal{L}\_{rw}$ for structural fidelity, a disentanglement loss $\mathcal{L}\_{dis}$ that soft-orthogonalises a node–subgraph affiliation matrix, and an entropy regulariser that avoids empty dimensions.
They also propose five new metrics—Comprehensibility, Sparsity, Overlap Consistency, Positional Coherence, and Plausibility—to quantify how human-understandable and disentangled the embeddings are.

**Audience:**

Yes

**Broader Impact Concerns:**

No ethical implications.

**Claims And Evidence:**

Yes

**Requested Changes:**

Provide ablation on $\lambda_{ent}$ and the number of encoder layers to show robustness.

**Strengths And Weaknesses:**

Strengths:
1. The paper is well-motivated. The paper persuasively argues that post-hoc explainers are insufficient for mission-critical applications, making intrinsic interpretability in node embeddings an important contribution.
2. The proposed loss for optimizing the interpretable embedding that blends random-walk preservation, soft orthogonality (for disentanglement) and an entropy term (to avoid empty factors) is conceptually simple yet effective.
3. The five proposed metrics fill the evaluation gap and move beyond purely accuracy-centred comparisons.

Weaknesses:
1. There's no ablation on $\lambda_{ent}$ or on GCN depth.
2. There lacks theoretical guarantees that the entropy + cosine objective yields disentangled factors; empirical evidence suffices but theory would strengthen claims.

---

> ### Author Response · Authors · 2025-05-14
> **Response to Reviewer j9WJ**
>
> We thank the reviewer for the insightful feedbacks.  Here we provide detailed answers to the specific concerns.
>
> While formal guarantees would undoubtedly add further rigor, our primary objective in this work is to demonstrate the empirical effectiveness of the proposed loss. We have therefore prioritized extensive experimental validation, which we believe provides strong evidence of its practical value. A comprehensive theoretical analysis, though beyond the scope of this paper, is a worthwhile direction that we intend to explore in future work.
>
> To complement our empirical focus, the new Figure A8 in the Appendix G of the revised paper investigates how the entropy-regularization coefficient $\lambda_{ent}$ and the GCN depth of DiSe-GAE affect interpretability metrics on the WIKI dataset. Comprehensibility (renamed Topological Alignment) and Overlap Consistency are positively affected by stronger regularization and deeper architectures. Sparsity decreases under the same conditions, highlighting a trade-off between compactness and explanatory power. Positional Coherence is only weakly influenced by $\lambda_{ent}$, but benefits from shallow networks. Moreover, it is worth highlighting that we achieved strong results in our main experiments without any extensive search of hyperparameters, beyond the embedding size, using the single setting $\lambda_{ent} = 1$ and $depth=1$. This suggests that the method is inherently robust and delivers competitive performance with minimal tuning.

---

### Review · Reviewer_EpKW · 2025-05-01

**Summary Of Contributions:**

The paper proposes a method, DiSeNE, for unsupervised node representation learning, that yields node embeddings that are both _disentangled_ and _self-explainable_.  The former implies that distinct embedding dimensions correspond to subgraphs that are in a precise sense minimally correlated.  The latter means that distinct embedding dimensions are associated with distinct subgraphs whose structures are encoded by those dimensions.

The paper additionally proposes several novel metrics for evaluating interpretable node representations and uses them to compare DiSeNE empirically with state of the art post-hoc explanation generators (e.g., GNNExplainer).

**Audience:**

Yes

**Claims And Evidence:**

Yes

**Requested Changes:**

1.) I recommend that "comprehensibility" be renamed in light of weakness #1.  Additionally, I wonder if the authors can discuss other notions of comprehensibility that measure the extent to which mask matrices capture other human-understandable structures.

**Strengths And Weaknesses:**

Strengths:

1.) Learning explainable node embeddings in an unsupervised setting is well-motivated, as it provides task-agnostic explainability.

2.) Interpretability is evaluated along several axes.

3.) Several real and synthetic datasets are used in the empirical evaluation.

Weaknesses:

1.) It is not clear that the evaluation metric called "comprehensibility" deserves the name.  Specifically, while community structure is an important structural characteristic of graphs, other structures may be important and human-understandable in some contexts.

---

> ### Author Response · Authors · 2025-05-14
> **Response to Reviewer EpKW**
>
> We appreciate the reviewer’s feedback regarding the naming and scope of the “Comprehensibility” metric. Our intent with this metric is to evaluate how well the explanation subgraphs produced by each embedding dimension align with one class of human-understandable structures: community modules. As communities are widely accepted as intuitive and semantically meaningful mesoscale structures in real-world graphs (e.g., social or biological networks), we used them as a proxy for human interpretability.
> That said, we agree that the term “comprehensibility” might be overly general, potentially implying a broader coverage of interpretable structures than what we currently assess.
> To address this, we renamed the metric to “Topological Alignment” to emphasize that it measures the alignment between embedding dimension explanations and ground-truth communities identified as topological structures.
>
> In Section 3.3 (revised version), we added a paragraph clarifying that this metric does not exhaustively capture all forms of comprehensibility, but rather focuses on communities as one prominent and well-defined class of human-understandable structure. Moreover, in the same paragraph, we also discuss potential extensions of our framework that could incorporate alternative ground-truth structural annotations, such domain-specific patterns in biological and social networks, to evaluate broader forms of comprehensibility.

---

### Author Response · Authors · 2025-05-14
**General response for all reviewers**

We sincerely thank the reviewers for their insightful comments, which have significantly contributed to improving our work. In response, we have submitted a revised version of the paper, and addressed all concerns through detailed point-by-point responses. All the additions in the revision are reported with blue characters. We strongly believe that these additions better support the proposed methodology, extending the initial submission with new empirical evidence.

Here we summarize the main changes of the uploaded revised paper:

- We updated Section 3.1 and afterwards by replacing the name **“Comprehensibility”** with **“Topological Alignment”**, following the suggestions from **R#EpKW**  about the related metric.

- In line with **R#dLkq**’s suggestion, we included an additional baseline method from the family of Stockastic Blockmodel-based GNNs, **DGLFRM [1]**. In the main paper, new results appear in Tables 1-2-3. In the Appendix, we have updated related  Figures and Tables with the new method.

- While incorporating the new scores, we uncovered a reporting error for **DiSe-GAE** and **DiSe-FCAE** in Table 2 (regarding exclusively the Overlap Consistency scores). The table has been corrected; this adjustment does not alter the outcome and discussion of this set of experiments.

- In the Appendix B, for answering **R#nkUJ**, we included Table A2 that compares the time and space complexity of our model against baseline methods.

- Following another suggestion of **R#nkUJ**, Appendix D now reports Figures A2-A3 which provide comprehensive results across multiple output embedding dimensions for real-world data. The same section extensively details a new experiment, summarized in Figures A4–A5 and Table A6, that goes deeper into understanding the accuracy-interpretability trade-offs of the approach in a more general setting.

- As requested by **R#j9WJ**, we added Appendix G with the new Figure A8 detailing ablation results for GCN depth and entropy regularization.

- In response to **R#dLkq**,  Appendix H now includes Tables A7-A8 which showcase the interpretability of CORA embeddings by relating structural patterns to topic and textual information when explaining the node classification task.

[1] Mehta, Nikhil, Lawrence Carin Duke, and Piyush Rai. "Stochastic blockmodels meet graph neural networks." ICML, 2019.

---

### Author Response · Authors · 2025-07-18
**Camera Ready Version Uploaded**

Dear AE, dear Reviewers,

We sincerely thank you again for the effort in managing and reviewing our submission. We would like to inform you that the camera-ready version of our manuscript has been uploaded.

Best Regards,

The Authors

---

### Decision · Action_Editor_M6W2 · 2025-06-26

**Recommendation:** Accept as is

**Audience:**

Yes

**Audience Explanation:**

Yes, the findings of this paper would be of interest to the TMLR audience, particularly those working on graph representation learning, interpretability, and unsupervised learning.

**Claims And Evidence:**

Yes

**Claims Explanation:**

Yes, the claims are supported by clear and convincing evidence. The proposed DiSeNE framework introduces novel objectives and evaluation metrics, and its effectiveness is demonstrated through extensive experiments on benchmark datasets in the original paper. After the rebuttal, the authors further enhanced the experimental evaluation by providing the time and space complexity analysis and other detailed ablation studies.